# Systemic Inflammation, Oxidative Stress and Cardiovascular Health in Children and Adolescents: A Systematic Review

**DOI:** 10.3390/antiox11050894

**Published:** 2022-04-30

**Authors:** Tjaša Hertiš Petek, Tadej Petek, Mirjam Močnik, Nataša Marčun Varda

**Affiliations:** 1Department of Paediatrics, University Medical Centre Maribor, Ljubljanska 5, 2000 Maribor, Slovenia; tjasa.hertispetek@ukc-mb.si (T.H.P.); mirjam.mocnik@ukc-mb.si (M.M.); natasa.marcunvarda@ukc-mb.si (N.M.V.); 2Faculty of Medicine, University of Maribor, Taborska ulica 8, 2000 Maribor, Slovenia

**Keywords:** oxidative stress, systemic inflammation, cardiovascular health, endothelial dysfunction, biomarkers, children and adolescents, antioxidants, exercise, diet, chronic disease

## Abstract

Recent studies indicate that cerebrovascular diseases and processes of atherosclerosis originate in the childhood era and are largely influenced by chronic inflammation. Some features of vascular dysfunction in adulthood may even be programmed prenatally via genetic influences and an unfavorable intrauterine milieu. Oxidative stress, defined by an imbalance between the production and generation of reactive oxygen species (ROS) in cells and tissues and the capability of an organism to scavenge these molecules via antioxidant mechanisms, has been linked to adverse cardiovascular health in adults, yet has not been systematically reviewed in the pediatric population. We performed a systematic search as per the PRISMA guidelines in PubMed/Medline and Cochrane Reviews and detected, in total, 1228 potentially eligible pediatric articles on systemic inflammation, oxidative stress, antioxidant use, cardiovascular disease and endothelial dysfunction. The abstracts and full-text manuscripts of these were screened for inclusion and exclusion criteria, and a total of 160 articles were included. The results indicate that systemic inflammation and oxidative stress influence cardiovascular health in many chronic pediatric conditions, including hypertension, obesity, diabetes mellitus types 1 and 2, chronic kidney disease, hyperlipidemia and obstructive sleep apnea. Exercise and diet may diminish ROS formation and enhance the total serum antioxidant capacity. Antioxidant supplementation may, in selected conditions, contribute to the diminution of the oxidative state and improve endothelial function; yet, in many areas, studies provide unsatisfactory results.

## 1. Systemic Inflammation and Oxidative Stress Influence Cardiovascular Health in Children and Adolescents

Cardiovascular diseases are the number one morbidity and mortality group of dis-eases in developed society and include coronary, cerebrovascular, aortic and peripheral vascular disorders. The common denominator, atherosclerosis, is a pathologic process defined as being, among other definitions, chronic inflammation. The inflammatory process is established through endothelial cell activation [1]. Endothelial cells are both the source and target of numerous factors contributing to atherosclerosis [2]. In fact, several hundred variables were shown to be associated with coronary disease [1].

In general, cytokines are known to be involved in several inflammation-related processes and must be regulated properly. However, their expression, production or activity are affected by several genetic and environmental factors [1,3], presented below. In healthy children, systemic inflammation and oxidative stress already influencing cardiovascular health seems to take place, which was demonstrated by a correlation between circulating inflammatory chemokines with vascular characteristics of the carotid artery and a positive association between C-reactive protein and oxidative stress [4,5].

The presence of an oxidized low-density lipoprotein and its deposition inside the arterial wall, recognition by macrophages and subsequent proinflammatory immune response is a major pathogenic mechanism of the atherosclerotic cascade [6]. Obesity and associated conditions perpetuate atherosclerosis not only among adults but also in children [2]. An increased cardiovascular risk has been demonstrated by elevated homocysteine and other cardiovascular parameters in overweight and obese children [7]. Additionally, systemic inflammation and increased intima-media thickness were interestingly associated with anti-food IgG antibodies in obese children [8].

It is evident that systemic inflammation and oxidative stress influence the cardiovascular system of children and young adults already at an early age and, with the introduction of oxidative stress biomarkers in pediatric medicine [9], should be researched in greater detail.

## 2. Materials and Methods

For this systematic review, we followed the recommendations stated in the Preferred Reporting Items for Systematic Reviews and Meta-Analyses (PRISMA) guidelines. We registered the systematic review protocol on PROSPERO. A PubMed/MEDLINE and Cochrane Reviews search was conducted by two investigators (T.H.P. and T.P.) up to the 7th of January 2022 for pediatric studies (keywords: infant *, child *, adolescent *, pediatric * or paediatric * or via filter: 0–18 years) reporting in the titles and/or abstracts on cardiovascular diseases (keywords: cardiovasc *, cardiometabol *, renal *, arterial *, endothel *, atheroscl *, hypertens * and “intima-media”) and systemic inflammation; oxidative stress or antioxidants (keywords: “systemic inflammation”, redox*, oxidative * and antioxida *). We used the asterisk wildcard character (*) to include alternative forms of words. A manual search of the relevant reference lists was also performed.

Both original research and review articles in English language were included. We searched for studies published between January 1992 and January 2022. Where possible, a filter was used to include only human studies in the ap-propriate age group. In the search, we excluded studies on oxidative stress in acute conditions (e.g., Kawasaki disease, Henoch–Schönlein purpura and COVID-19-related “multisystem inflammatory syndrome in children”), which have been already reviewed [10,11] and are briefly discussed at the end of the article. We also excluded studies on oxidative stress in diseases of prematurity (e.g., bronchopulmonary dysplasia and hypoxic–ischemic brain injury) or other studies not directly related to cardiovascular health (e.g., oxidative stress in the autism spectrum disorder, markers of oxidative stress in children undergoing heart surgery, etc.). Letters to the Editor, conference proceedings and study protocols were deemed not eligible for this review. Any discrepancies regarding the inclusion of a study were discussed with the supervisor (N.M.V.) and resolved. The included studies were further assessed for the quality of their design and of the results presented. Both positive and negative studies were included. In total, we screened 1228 titles and abstracts and assessed 1186 full-text reports for eligibility. One hundred and sixty studies met the inclusion/exclusion criteria and are presented below. A list of reported studies with study characteristics and main findings is reported in Table 1. Table 2 summarizes the risk factors and diseases associated with increased oxidative stress and systemic inflammation, and Table 3 lists the oxidative and inflammatory markers associated with cardiovascular disease, as reported in the manuscript.

## 3. Genetic and Perinatal Factors Affect Oxidative Stress and Systemic Inflammation

Though the timing for the onset of vascular alterations is unknown, some investigations indicate differences between vascular function already prenatally. It is sometimes difficult to determine if this is a consequence of genetic or perinatal factors [12]. For example, overweight and obesity during pregnancy have been associated with increased birth weight, childhood obesity and noncommunicable diseases in the offspring, whereas the extent of the prenatal, genetic and postnatal environmental and behavioral effects has yet to be elucidated [145]. Similarly, ultrasound studies of neonatal and fetal aorta indicate that impaired fetal growth, in utero exposure to maternal hypercholesterolemia and diabetic macrosomia present important risk factors for vascular changes consistent with the early signs of atherosclerosis [146]. (Figure 1).

The expression of inflammation-related molecules of endothelial cells in healthy neonates with a strong family history of myocardial infarction was more pronounced than in neonates without a positive family history [12]. Additionally, children of hypercholesterolemic mothers had different arterial gene expressions supporting the assumption of genetic programming in utero [12]. Similarly, according to the fetal programming hypothesis, malnourished neonates exposed to placental insufficiency exhibit endothelial cell dysfunction. There is growing evidence that being small for their gestational age presents a risk factor for developing type 2 diabetes, metabolic syndrome and cardiovascular disease in adulthood [147].

The loss of appropriate nitric oxide production or activity seems to be a cornerstone for appropriate endothelial function, which is lost during the mitochondrial damage in malnourished fetuses [147]. Additionally, reactive oxygen species are generated in the presence of fetal hypoxia, creating oxidative stress, further contributing to endothelial cell dysfunction, vascular smooth muscle cell proliferation and apoptosis [147,148]. Preterm infants are particularly susceptible to the damaging effect of oxidative stress, and neonatal hyperoxia exposure leads to vascular dysfunction, hypertension, microvascular rarefaction and a reduced nephron number [149]. Kidney development is affected by hyperoxia, with mildly increased renal tubular necrosis, dilatation, regeneration and interstitial inflammation. Oxidative stress is also implicated as an important molecular mechanism in the initiation and progression of fibrosis in a variety of organs, including the kidneys, liver and lungs [150].

The consequences of altered fetal programming due to an unfavorable prenatal environment have also been demonstrated later in life in children before puberty with elevated markers of inflammation [13]. Preventive approaches such as breastfeeding, supplementation with folate, vitamins, antioxidants, L-citrulline, L-arginine and treatment with nitric oxide modulators represent promising strategies for improving endothelial function, mitigating long-term outcomes and possibly preventing intrauterine growth restriction of vascular origin. The identification of early biomarkers of endothelial dysfunction could allow the early screening and follow-up of individuals at risk of developing cardiovascular and renal diseases [151].

Interestingly, children conceived via classic in vitro fertilization had significantly higher systolic and diastolic blood pressures and triglycerides; however, no evidence of insulin resistance, adipokines and inflammatory markers was detected [14].

Several clusters of cardiovascular diseases in the families indicate a genetic background of cardiovascular risk development besides sharing similar environmental and behavioral factors. Positive parental history of premature coronary heart disease was as-sociated with inflammatory factors and markers of oxidation, carotid intima-media thickness and left ventricular mass [15,16]. This cardiovascular risk manifests already at an early age, demonstrated by at-risk boys 6–8 years of age (based on maternal cardio-vascular health and lifestyle habits) with increased markers of oxidative stress, arterial stiffness and diastolic blood pressure [17]. A positive family history for coronary heart dis-ease also increases the risk for congestive heart failure and myocardial infarction death at an early age. Likewise, inheritance patterns in both cerebral and aortic aneurysms have been demonstrated [12].

The genetic background is challenging to decipher and still subject to ongoing re-search. Some polymorphisms have been associated with blood pressure, carotid intima-media thickness, the prooxidant status and oxidative stress indicators [18,19,20]. Excessive oxygen-free radical production due to the imbalance between pro-oxidation and antioxidation seems to develop early in the pathogenesis of hypertension [21], as does intensive lipid peroxidation [22]. Polymorphisms of enzymes in antioxidant pathways have been linked to heart disease, such as, for example, a polymorphism in the glutamate-cysteine ligase gene with coronary dysfunction and myocardial infarction [166].

Ethnic differences support genetic backgrounds of cardiovascular risk development. Differences in the distribution of risk factors and disease between race and ethnic groups are a function of the frequency of specific genotypes and interactions with environmental factors. The most important differences include higher blood pressure, lower triglycerides and higher high-density cholesterol in the Black race, a higher prevalence of diabetes and insulin resistance among Hispanics and higher triglyceride levels among the Japanese compared to Caucasians [167].

## 4. Effect of Selected Chronic Diseases on Cardiovascular Health via Oxidative Stress and Systemic Inflammation

### 4.1. Hypertension and Vascular Dysfunction

Essential hypertension is a complex disease, contributing to the acceleration of atherosclerosis and cardiovascular diseases, originating from genetic predisposition and environmental factors. Hypertension origin may even start prenatally with reactive oxygen species, generated due to the exposure of adverse in utero conditions, causing the developmental programming of hypertension [152].

Several studies have indicated numerous factors in hypertension development. For example, a hypothesis present in the literature assumes that essential hypertension belongs to a group of psychosomatic diseases, where the emotional status affects the neurohumoral mechanisms of regulation of the cardiovascular system. Repeated exposure to negative psycho-emotional factors leads to increased anxiety, irritability and, predisposition to depressive reactions, causing a permanent overstrain in the sympathoadrenal system, followed by a spasm of smooth muscles in arterioles. In adolescents with essential hypertension, pronounced changes in their emotional status associated with dysregulation in the lipid hydroperoxide antioxidant protection system were demonstrated [23].

Interestingly, dental caries were associated with primary hypertension development in children. The rationale was that dental caries would increase the risk of systemic inflammation in a variety of ways. It may be explained by prolonged cariogenic bacteria stimulation, increasing inflammatory mediators’ contributions to endothelial dysfunction and artery plaque formation. Among hypertensive patients with poor oral status, the intensive oxidation of several plasma substrates, increase in reactive metabolites of oxygen, lipid peroxidation, inactivation of prostacyclin and nitric oxide and an imbalance in the total antioxidant capacity were noted [24].

From a genetics point of view, several candidate genes for essential hypertension development are present. In adults, endothelial nitric oxide synthase gene polymorphism was associated with the risk of essential hypertension and vascular complications. How-ever, in children with hypertension, the altered allele was not more prevalent but was as-sociated with earlier vascular damage [25]. Decreased nitric oxide bioavailability is associated with atherogenesis and can participate in enhanced cell adhesion, proliferation, vasoconstriction and the generation of atherosclerotic lesions, explaining earlier vascular damage in the gene alteration [153]. In addition, in children, gene polymorphisms of the renin–angiotensin–aldosterone system and aldosterone synthase genes were proposed as risk factors. A higher risk of hypertension was demonstrated also with the single-nucleotide polymorphism of ATP2B1 (ATPase Plasma Membrane Ca^2+^ Transporting 1) [154]. Another interesting candidate gene is paraoxonase 1 (PON1), associated with high-density lipoproteins in plasma, capable of the hydrolysis of oxidized lipids and prevention of the oxidation of low-density lipoproteins. However, in adolescents with essential hypertension, there were no significant differences in the genotype distributions and the allele polymorphism frequencies [26].

Low-grade inflammation plays a role in the pathogenesis of essential hypertension, where vascular inflammation precedes systemic inflammatory changes [27]. The presence of systemic oxidative stress was proven in hypertensive children and adolescents irrespective of their body mass index, with decreased levels of nitrates and increased levels of lipid peroxidation end products. The ratio between lipid peroxidation and nitric oxide is correlated directly with both systolic and diastolic blood pressures for the overall patient population [28]. A state of oxidative stress is thought to be associated with adipocytokine release, associated with obesity and metabolic syndrome and renin–angiotensin–aldosterone system activation. According to the research, a state of oxidative stress correlates with organ damage (left ventricular hypertrophy and carotid intima-media thickness) in hypertensive children, along with metabolic abnormalities, fat amount and insulin resistance [29]. Additionally, noninvasive urinary biomarkers of oxidative stress have been shown to correlate with an arteriosclerosis index in school children and may predict the risk of developing lifestyle-related diseases [30].

Antioxidant mechanisms are also being studied, such as thiol/disulphide homeostasis. Thiols play a critical role in preventing the formation of oxidative stress in cells. They are converted into reversible disulphide structures in the case of oxidative stress, and disulphide bonds are again reduced into thiol groups when oxidative stress resolves. Higher disulphide levels therefore indicate increased oxidative stress and were demonstrated in adolescents with essential hypertension [31]. Additionally, in children with hypertension, a reduced antioxidative capacity by significant glutathione depletion was shown compared to the body mass index of matched controls [28].

### 4.2. Obesity

Several chronic diseases exacerbate oxidative stress and systemic inflammation. Obesity, which has reached epidemic proportions in children in the last few decades, is associated with the development of other cardiovascular risk factors. Obesity by itself is associated with a proinflammatory and prothrombotic state. The latter was demonstrated by altered coagulation results (fibrinogen, D-dimer, prothrombin time, endogenous thrombin potential and von Willebrand factor) in obese children and adolescents [32,33]. The question remains how to assess the cardiovascular risk and whether these abnormalities are reversible with earlier interventions [34].

Obesity-related hypertension is becoming an important cardiovascular risk in children, developed by alterations in obesity with endocrine determinants, such as cortico-steroids and adipokines, sympathetic nervous system activity and disturbed sodium homeo-stasis, as well as oxidative stress, inflammation and endothelial dysfunction [155]. A systemic oxidative status was associated with systolic blood pressure and pulse pressure in children with obesity [35]. Additionally, T-helper cells could be activated in obese hypertensive children before the onset of clinical indicators of target organ damage [36]. In children with obesity, low-grade systemic inflammation and endothelial activation were independent of blood pressure modulation and may have an influence on elevated blood pressure relatively early in life [37].

Excess fat in children is also associated with an increased risk for developing dyslipidemia, diabetes mellitus, hepatic cholestasis and metabolic syndrome [38,39,40,41]. Metabolic syndrome in children has been associated with a higher incidence of cardiovascular disease and all-cause mortality during adulthood [39], making it a priority to treat obesity early. Waist circumference measurement is an important anthropometric parameter in children with obesity and metabolic syndrome, easily obtained during a physical examination and could predict increased cardiovascular risk [42].

The determination of obesity’s effect on atherosclerosis alterations and further complications is a major challenge. Obesity is believed to be a state of increased oxidative stress [168], yet is difficult to measure. Reliable markers to evaluate oxidative stress and inflammation in obesity are still not known, but numerous markers are under investigation. In recent years, the measurements of protein and lipid oxidation products are taking place to determine their use in clinical settings [43]. The oxidized low-density lipoprotein concentration was elevated in children with obesity before the carotid intima-media thickness increased [44]. Similarly, antioxidants are being investigated, such as thiol/disulphide homeostasis, which was impaired in obesity, indicating its contribution to oxidative stress and inflammation in obesity [45]. Often, nitric oxide is being investigated. It was increased in relation to fat accumulation and translated into higher values of cardiometabolic risk markers in children [46]. Polyamines, derived from arginine (precursor of nitric oxide), are being investigated and found to be significantly higher in obese children [47]. Chemerin, a chemoattractant protein, has been shown to be expressed in adipose tissue, suggesting an association between obesity and inflammatory and endothelial activation markers, and supports the role of chemerin as a molecular link between an increasing fat mass and an early atherogenic risk profile in obesity [48,49]. It was also associated with increased systolic pressure in obese children [50].

Catestatin, a lesser-known peptide with a wide spectrum of biological activity, such as inhibition of the catecholamine release, degree of blood pressure, stimulation of histamine release, reduction of beta-adrenergic stimulation and regulation of oxidative stress, was significantly lowered in children with obesity [51]. The stromal-derived factor and soluble E-selectin are possible indicators of the beginning of insulin resistance and endothelial damage [42]. Adipokines, including leptin, adiponectin, resistin and visfatin, secreted from adipose tissue and elevated in obesity, as well as cytokines and chemokines, contribute to the pathophysiology of obesity-related disorders [52], especially to type 2 diabetes mellitus development [53]. Leptin, the best known adipokine, was associated predictively with interleukin-6 (IL-6) in youths and was associated with the grade of overweight. Higher IL-6 levels were also detected in young children with not only obesity but fully developed metabolic syndrome [54]. IL-6 was also higher in children with type 2 diabetes mellitus [53].

Adiponectin is the most abundant serum adipokine, mainly secreted from white adipose tissue, and has demonstrated antiatherogenic, anti-inflammatory and insulin-sensitizing effects. Low levels of adiponectin were associated with the development of cardiovascular complications of obesity and were associated with cardiovascular disease even in children and adolescents [169]. However, recent studies also indicate its proinflammatory roles in patients with chronic diseases, e.g., chronic kidney disease. The bilateral pro- and anti-inflammatory functions of adiponectin presumably originate in different protein isoforms [170].

Insulin-like growth factor binding protein-3 was reduced in obesity, as well as its expression in macrophages which led to the suppression of its anti-inflammatory function and is believed to be an early marker of atherosclerosis [55]. Elevated urine 8-isoprostane was linked to the body mass index, waist circumference and ambulatory blood pressure [56]. Along with 8-isoprostane, C-reactive protein can be detected in urine, and both could serve as a noninvasive marker for the early detection of cardiovascular risk [30,57].

Other factors associated with obesity are potential contributors for vascular complications in obese children. In obese adults, it has been shown that acute glucose consumption induces a transient impairment in endothelial function and an increment of inflammation and oxidative stress, which was not present in obese youth. However, the association might have implications for individuals with impaired glucose tolerance or type 2 diabetes mellitus [58]. Hypovitaminosis D is frequently associated with obesity, commonly because of a sedentary lifestyle; however, in children with obesity and vitamin D insufficiency, increased markers of oxidative stress, inflammation and endothelial activation were demonstrated [59,156]. Vitamin D is also prominent in modulating the innate immune response to different pathogens and regulating the adaptive immune response in inflammatory and autoimmune disease. Vitamin D inhibits the production of IL-6 and tumor necrosis factor-alpha (TNF-α) and reduces the expression of monocyte chemoattractant protein-1 [156]. Despite this, the supplementation of vitamin D in children with obesity or overweight did not affect the measures of arterial endothelial function or stiffness, systemic inflammation or the lipid profile but resulted in reductions of blood pressure and the fasting glucose concentration, as well as in improvements in insulin sensitivity [60]. Similarly, serum vitamin B12 concentrations were negatively associated with proinflammatory cytokines and biochemical markers of cardiometabolic risk in adults but not necessarily associated with obesity [61]. Hyperhomocysteinemia in children with obesity was associated with a lack of homeostatic regulation with the elevation of proinflammatory chemokines, implicated in the initial stages of the inflammatory part of the atherosclerotic process [62].

Interestingly, not only obesity but also underweight has been associated with in-creased carotid intima-media thickness, oxidative stress, impaired inflammation and insulin sensitivity, indicating that impaired adipocytes stores (reduced or elevated) seem to result in a similar impaired endothelial dysfunction and early sign of accelerated atherosclerosis [63]. Furthermore, a behavioral intervention resulted in a paradoxical increase in some biomarkers in children. However, the study was limited by the number of participants and biomarkers used [64]. Another study, on the contrary, showed decreased chemerin and lipopolysaccharide-binding protein in association with metabolic risk factors after a lifestyle intervention program [65]. Furthermore, oxidative stress was decreased by improving the antioxidant defenses through fat volume reduction [157].

### 4.3. Metabolic Syndrome and Type 2 Diabetes Mellitus

Obesity, discussed above, is a central pathological mechanism in metabolic syn-drome development, which consists of pro-atherosclerotic metabolic abnormalities, including elevated blood pressure, blood glucose, waist circumference and triglycerides, as well as lower levels of high-density lipoproteins [158]. The etiology is multifactorial and includes a genetic background with environmental risk factors. Anyone who has a metabolic syndrome is more likely to have a family member with its components. The presence of metabolic syndrome in parents is predictive of subclinical inflammation in children that may be associated with the development of atherosclerotic disease in the future [66]. It is now well-recognized that adipose tissue, especially visceral fat, is not merely energy storage but an active endocrine organ that produces many bioactive molecules. In the setting of obesity, the overproduction of proinflammatory and prothrombotic adipokines is associated with inflammation, which has a central role in the pathogenesis of metabolic syndrome development and mediates its impact on cardiovascular diseases [158].

Key adipokines include interleukins (interleukin-1, -6, -10 and -18); adiponectin; resistin; tumor necrosis factor alpha; leptin; monocyte chemoattractant protein-1; angiotensinogen; plasminogen activator-inhibitor-1; myeloperoxidase and E-selectin [67,158]. Additionally, ceruloplasmin and 8-isoprostane might be useful tools in identifying patients with the highest risk of future cardiovascular disease [68,69]. The cooccurrence of metabolic syndrome and elevated inflammation markers was associated with a greater increase in arterial stiffness and carotid intima-media thickness, both measures of vascular changes in atherosclerosis [70].

Bilirubin might also play a role in metabolic syndrome development. Along with its involvement in biliary and hematologic systems, it has a potent antioxidant and cytoprotective function, allowing it to inhibit multiple steps in the formation of atherosclerosis. It has been associated with carotid intima-media thickness and ischemic cardiac disease. The serum total bilirubin levels were inversely correlated with the prevalence of metabolic syndrome, and the mechanism of the association might be related to insulin resistance [71].

Some of the newer biomarkers also include carotenes and tocopherols and dietary vitamins with antioxidant properties, which were lower in metabolically unhealthy children. In the same study, the plasma total antioxidant capacity was higher in pubertal children, supporting the importance of considering both the antioxidant and oxidative stress status and puberty in metabolic syndrome [72].

Diabetes mellitus type 2 is a common comorbidity of obesity and metabolic syn-drome. In the last few decades, the burden of the disease has been increasing in the pediatric population. Understanding the pathways involved in the inflammatory and vascular complications that accompany these comorbidities is one of the main focuses in this field. In this regard, irisin might play a pathophysiological role. It is a novel adipomyokine, secreted mainly by skeletal muscles following acute bouts of exercise and, to a lesser extent, by adipose tissue. The serum irisin levels were significantly lower in children with metabolic syndrome or type 2 diabetes mellitus, with negative correlations between irisin and the body mass index percentile. Lower irisin levels could therefore induce a lack of inhibition of oxidative stress and inflammation [73].

### 4.4. Hyperlipidemia

Familial hypercholesterolemia, an inherited disorder of the lipoprotein metabolism, is a well-described independent risk factor for premature atherosclerotic disease. Even asymptomatic patients with moderate and severe hypercholesterolemia have evidence of oxidant stress from a demonstration of lipid peroxidation with an increment of F_2_ isoprostanes [74]. In children with familial hypercholesterolemia, inflammatory and hemostatic abnormalities were present, namely plasminogen activator-inhibitor-1, interleukin-1β, intracellular cell adhesion molecules, endothelium-dependent reactive hyperemia and endothelium-independent nitrate hyperemia dilatation, indicating an inflammatory pathophysiological rationale to endothelial dysfunction and atherosclerosis [75]. Specifically, inflammation could be mediated via the involvement of monocyte-derived RANTES (Regulated upon Activation, Normal T Cell Expressed and Presumably Secreted), which is a chemokine and chemoattractant for monocytes and memory T-helper cells, as well as eosinophils. Children with familial hypercholesterolemia showed a significantly higher gene expression of chemokines, and specifically, higher levels of RANTES were present [76].

Additionally, oxidative stress might play a significant role in children with obesity-associated hypercholesterolemia, demonstrated by elevated levels of nicotinamide-adenine dinucleotide phosphate oxidase, along with the oxidized low-density lipoprotein levels, compared to a group of healthy children, children with only obesity or children with only familial hypercholesterolemia. The association of multiple cardiovascular risk factors and nicotinamide-adenine dinucleotide phosphate oxidase is related to a greater endothelial dysfunction and to enhanced oxidative stress in children [77,78].

### 4.5. Chronic Kidney Disease and Dialysis

The kidneys play a key role in whole body fluid and electrolyte homeostasis and, hence, in long-term regulation of the arterial pressure [159]. Chronic kidney disease represents a large group of kidney diseases that might play a role in cardiovascular disease and is being increasingly recognized as a novel cardiovascular risk factor [79]. Low levels of serum albumin and high levels of uremic metabolites might be responsible for increased oxidative stress [80]. Another possible explanation is cellular hypoxia, especially in patients needing renal replacement therapy [81]. An increased oxidative stress in T-lymphocytes has been demonstrated in children with end stage renal disease [82]. The precise mechanism of renal disease-induced oxidative stress has therefore not been completely explained; however, growing evidence suggests an association between chronic kidney disease and a state of chronic inflammation [79]. Cytokines, released in chronic inflammation, play an important role in modulating renal hemodynamics and cardiovascular responses [159]. Interestingly, some antioxidant and oxidant markers were also significantly elevated in the saliva of children with chronic kidney disease, demonstrating the effect of oxidative stress in this group of patients, along with possible noninvasive methods of oxidative stress evaluation [83].

End stage renal disease with the need for renal replacement therapy is a well-recognized cardiovascular risk factor, and it is believed to be a state of oxidation with imbalance between pro- and antioxidants. This was demonstrated by increased inflammatory markers and reduced plasma glutathione, an important antioxidant, in children with end stage renal disease and on hemodialysis [84,85,86]. Hemodialysis itself leads to an increased loss of antioxidants and antioxidant vitamins, leading to a reduced activity of the antioxidant defense system and contributing significantly to increased oxidative stress [81,87]. Even more, it has been shown that oxidative stress is aggravated during every single hemodialysis session [88]. In this group of patients, oxidative stress was correlated with the degree of cardiac dysfunction [84]. Inflammatory markers were also associated with left ventricular hypertrophy [89,90]. Additionally, children with chronic renal failure, not necessarily on dialysis, demonstrated similar oxidative stress and inflammation biomarkers, together with the early cardiovascular damage presented with an increased left ventricular mass of the heart and increased carotid intima-media thickness [79,91]. Dyslipidemia contributes significantly to early atherosclerosis in children with chronic kidney disease [92,93]. Vascular or valvular calcifications indicate a poor prognosis in terms of the overall survival and cardiovascular morbidity and mortality. They are more common in populations over the age of 65 years, with a prevalence of two percent [160]. Additionally, in children requiring hemodialysis for several years, coronary calcifications are common. Worse renal osteodystrophy control and malnutrition with low cholesterol may be contributing factors to the formation of coronary calcifications [94].

In children after kidney transplantation, hypertension presents a significant risk of oxidative stress-induced organ damage with possible renal graft damage and should be addressed properly. Cyclosporine or tacrolimus are commonly used after transplantation, both further increasing the blood pressure. It is believed that they hamper the nitric oxide bioavailability and increase the reactive oxygen species, and therefore, immunosuppressive treatment significantly induces oxidative stress-related processes and is related to posttransplant hypertension [95].

One of common conditions in children is hydronephrosis, developed by obstruction at the level of the pelvo–ureteric junction due to the abnormal development. Preoperatively, the mean arterial pressure was significantly higher in hydronephrotic patients compared to healthy controls, with a reduction after surgical correction. Additionally, the markers of oxidative stress were significantly increased in patients with hydronephrosis and were again reduced following surgery. A study also found a trend for increased nitric oxide synthase activity and a signaling mechanism, which might be a compensatory mechanism [96]. Increased oxidative stress was also demonstrated in glomerulonephritis, pyelonephritis and lower urinary tract infections [97].

### 4.6. Obstructive Sleep Apnea

Obstructive sleep apnea is a continuum of sleep-disordered breathing of severity from partial obstruction of the upper airway producing snoring to increased upper airway resistance syndrome and to continuous episodes of complete upper airway obstruction. Repeated episodes of upper airways obstruction during sleep lead to significant hypoxemia and to cyclical alterations of arterial oxygen saturation with oxygen desaturation developing in response to apnea, followed by the resumption of oxygen saturation during hyperventilation leading to a phenomenon called hypoxia/reoxygenation. This may alter the oxidative balance through the induction of excess oxygen-free radicals. Increased levels of systemic biomarkers of inflammation and oxidative stress have been frequently demonstrated, suggesting a possible role in the pathogenesis of atherosclerosis [98,99,100]. Higher levels of IL-6 and 8-isoprostane were shown in exhaled breath condensate in children with obstructive sleep apnea, along with a positive correlation with the degree of cardiac dysfunction [98]. NOX2, the catalytic core of nicotinamide adenine dinucleotide phosphate oxidase (the most important source of cellular superoxide anion production and a source of reactive oxygen species) was significantly elevated in children with obstructive sleep apnea with a decreased flow-mediated dilatation [101]. In contrast, obstructive sleep apnea in children was not directly associated with structural and functional carotid changes [100]. Sleep-disordered breathing is more prevalent in children with obesity, further affecting accelerated atherosclerosis [102].

### 4.7. Type 1 Diabetes Mellitus

Type 1 diabetes mellitus is an autoimmune disease, heralded by anti-beta cell antibody formation and subsequent beta-cell destruction, associated with a state of chronic hyperglycemia that produces high levels of advanced glycosylation end products. The latter cause the expression of intracellular adhesion molecules and E-selectin on vascular endothelial cells that help the binding of macrophages and other inflammatory cells with the transendothelial migration of white blood cells into subendothelial spaces. Therefore, type 1 diabetes mellitus causes endothelial dysfunction and early atherosclerosis, which can result in premature cardiovascular events, as well as micro- and macrovascular complications [103,104].

Several proinflammatory markers are elevated prior to the development of the arterial disease [105]. Simultaneously, nitric oxide overproduction and decreased antioxidative protection is evident, especially in children with poorly controlled type 1 diabetes. Changes were detected even in the first year of the disease but were more pronounced later in disease progression, further contributing to later vascular complications [106]. Plasma E-selectin was also significantly higher with elevated HbA1c, which correlated with the carotid intima-media thickness and peripheral arterial tonometry, reflecting vascular damage [103]. In another study, the mean carotid intima-media thickness was higher in children with type 1 diabetes mellitus and strongly associated with the total cholesterol, low-density cholesterol, length of the disease, positive family history of diabetes and early cardiovascular events but not to the oxidative stress parameters [107].

Poorly controlled type 1 diabetes mellitus is a more pronounced significant risk factor for accelerated atherosclerosis and vascular complications, as there was an important relationship between HbA1c and oxidative stress [108]. This fact emphasizes the need for good glycemic and inflammatory follow-up to secure on-time interventions. Novel biomarkers are being investigated, such as urinary α-tocopherol, a vitamin E metabolite, which was significantly elevated in children with type 1 diabetes mellitus [109].

The increased levels of high-density lipoproteins are commonly elevated in children with type 1 diabetes mellitus. However, their cardiovascular-protective effect seems to transform into a dysfunctional proinflammatory equivalent in the presence of a chronic disease. Namely, in children with type 1 diabetes mellitus and chronic kidney disease, increased levels of high-density lipoproteins were observed, along with the elevation of chronic inflammation markers [110].

## 5. Influence of Exercise and Diet on Oxidative Stress, Inflammation and Cardiovascular Health in Children and Adolescents

In children at risk for subclinical atherosclerosis due to inflammation and oxidative stress, early interventions targeting microvascular health might help safeguard against future cardiovascular and neurodegenerative diseases. Exercise, optimally with an appropriate diet, is known to improve arteriolar dilation, increase insulin sensitivity and reduce the systemic low-grade inflammation that accompanies cardiovascular disease [111,112]. Additionally, pharmacologic treatment and bariatric surgery present treatment options but are not frequently used in the pediatric population. They are no substitute for dietary and lifestyle interventions. Even in children receiving cholesterol-lowering medications, such as statins, physicians should take every opportunity to encourage children and their parents to follow healthy diet and lifestyle choices. The same is true for bariatric surgery, which usually leads to significant weight loss. However, glycemic control might not always be improved, especially in diabetes mellitus type 1 [171,172].

Exercise enhances the antioxidant capacity of the body, leading to a reduced generation of reactive oxygen species both at rest and in response to exercise stress. These responses seem to be affected by factors such as the training phase, training load, fitness level, mode of exercise and so on [161]. Reduced oxidative stress markers in children with more frequent physical activity with less sedentary time may diminish the need for maintaining high concentrations of antioxidants in plasma during rest to achieve redox homeostasis [113]. Interventions to improve physical fitness, insulin sensitivity and reduce inflammation and endothelial dysfunction are already studied in the pediatric population and present a group of patients that likely benefit from the interventions the most, reducing cardiovascular disease and improving cerebrovascular function later in life [111,162].

Several studies have confirmed the positive effect of exercise in overweight children and adolescents, with improvement of fitness, body mass index, serum lipids, markers of oxidative stress and endothelial function [114,115]. Furthermore, physical activity was inversely related to some inflammatory markers independent of adiposity and fat localization [116]. In normal weight children and adolescents, physical activity increases nitric oxide bioavailability [115]. Exercise was also beneficial in the cases of novel markers, such as chemerin, which decreased after intervention, along with improvements in glucose and lipid metabolism [117]. On the contrary, the isoprostane levels were related to several markers of cardiovascular risk at the baseline; however, despite reduced fatness and improved fitness, no effect from exercise was observed in the isoprostane levels [118]. An exercise intervention program also demonstrated improved vascular elasticity, assessed by the pulse wave velocity in obese youth after weight reduction [119]. In children with obesity, the best way to reduce the fat mass and improve vascular stiffness is regular training that combines both muscle strengthening and aerobic components. Long-term adherence is of utmost importance [173].

Furthermore, nutrition might play an additional role in low-grade inflammation, contributing to the pathogenesis of atherosclerosis and, also, to cognitive function. Sometimes, these foods are therefore called “brain food”. Specifically, the polyunsaturated omega-3 fatty acids intake has beneficial effects on human health, especially on cardiovascular disease, via the decreased production of inflammatory eicosanoids, cytokines, reactive oxygen species and the expression of adhesion molecules. They include eicosapentaenoic acid and docosahexaenoic acid, found mainly in oily fish and fish oils, and alpha-linolenic acid (the precursor of the first two listed), found principally in walnuts [174]. Interestingly, a carbohydrate-restrictive (ketogenic) diet has been linked to decreased oxidative stress and improvements in the mitochondrial respiratory complex activity [175]. A specific “methionine-restrictive” diet has also been described with improvements in the lipid metabolism, decreased systemic inflammation and increased oxidative capacity without necessitating caloric restriction [176].

A lack of dietary antioxidant intake was associated with an adverse cardiometabolic profile in children and adolescents. A high prevalence of carotene deficiency was found in children with obesity or a metabolically unhealthy status [72]. Similarly, the low total antioxidant dietary intake was evaluated as a potential risk factor for the development of obesity-related features [177]. Last, but not least, nutritionally stunted children showed increased oxidative stress and a decreased antioxidant defense system [178], highlighting the importance of the dietary antioxidant intake.

## 6. Use of Antioxidants and Their Effects on Cardiovascular Health

### 6.1. Vitamins, Minerals and Coenzymes

Children with cardiovascular risk have increased oxidative stress and decreased antioxidant defense, as noted above. Most often mentioned, impaired antioxidants include reduced endogenous levels of vitamins C and E and reduced glutathione and increased levels of malondialdehyde and oxidized low-density lipoproteins, as well as decreased levels of superoxide dismutase and a decreased total antioxidant capacity [120]. Therefore, therapy with a supplementation of antioxidants may be in place; however, the studies are not conclusive about their usefulness.

Vitamin C and E supplementation did not decrease oxidative stress, improve endothelial function or increase the vascular repair capacity in patients with type 1 diabetes mellitus [120]. Although therapy with vitamin C did not alter the endothelial function in asymptomatic patients after the Fontan procedure, it provided some benefits in a subgroup of patients with abnormal vascular function [121]. Additionally, treatment with vitamin C blocked the acute hyperglycemic impairment of endothelial function in adolescents with type 1 diabetes mellitus [122]. Supplementation with vitamin E in adolescents with type 1 diabetes mellitus and early signs of retinopathy did not modify the pathological vascular process [123]. However, it improved the therapeutic effect of recombinant human erythropoietin in children with chronic renal failure on hemodialysis [124].

Discoveries of new functions for vitamin K-dependent proteins in the last decades have led to substantial revision of the vitamin K physiology. Vitamin K-dependent proteins are now known to be present in virtually every tissue and are important in bone mineralization, arterial calcification, apoptosis, phagocytosis, growth control, chemotaxis and signal transduction. As regards atherosclerosis, the dietary levels of vitamin K might be involved in arterial calcification [179].

Zinc is an essential trace element and a component of many enzymes. It is involved in the synthesis, storage and release of insulin. It has been documented that a zinc deficiency may be a predisposing factor for insulin resistance, glucose intolerance and diabetes mellitus, as well as atherosclerosis and coronary artery disease. Zinc supplementation contributed to improve the apolipoprotein A1 and B levels, oxidized low-density lipoprotein, leptin, malondialdehyde and C-reactive protein. The markers of insulin resistance also decreased significantly [125].

Similarly, coenzyme Q10 has been studied, as it is the coenzyme of the mitochondrial enzyme complexes involved in oxidative phosphorylation in the production of adenosine triphosphate and natural antioxidants. However, supplementation with coenzyme Q10 in children with chronic heart failure due to dilatative cardiomyopathy did not result in significant improvements, as well as no favorable effects on endothelial dysfunction or the metabolic parameters in patients with type 1 diabetes mellitus [126,127].

### 6.2. L-Arginine

L-arginine has been used to improve endothelial dysfunction by increasing the nitric oxide bioavailability in animal models; however, data in human research is scarce and contradicting. In children with chronic renal failure, supplementation with L-arginine did not improve endothelial function [128]. On the other hand, L-arginine supplementation in hypertensive patients after cardiac transplantation reversed endothelial dysfunction and attenuated high blood pressure [129].

### 6.3. Melatonin

Melatonin is an endogenously produced molecule secreted by the pineal gland and has multiple functions, some of which have antioxidant and anti-inflammatory effects. In adults, it has been shown to be of use for cancer, neurodegenerative disorders and aging. In children and neonates, melatonin has been used widely, including for respiratory distress syndrome, bronchopulmonary dysplasia, periventricular leukomalacia, hypoxic ischemic encephalopathy and sepsis. In addition, melatonin can be used in childhood sleep and seizure disorders and in neonates and children receiving surgery [163]. It is also present in breast milk and could influence children while breastfeeding. Melatonin can also participate in the gut microbiota composition. In all these ways, melatonin from breast milk influences weight gain in infants, limiting the development of obesity and comorbidities in the long term, and can influence the cellular environment for the development of infants’ cardiovascular systems [164].

### 6.4. Flavonoids and Natural Antioxidants

Natural antioxidants and traditional Chinese medicines are becoming increasingly popular with the increase of cardiovascular burden in general. Flavonoids, extracted from *Bidens bipinnata*, are traditionally known in Chinese medicine for their antipyretic, anti-inflammatory and antirheumatic effects via inhibition of the production of inflammatory cytokines. Henoch–Schönlein purpura manifests as the systemic inflammation of small vessels presenting in 20−55% of patients with hematuria, proteinuria and hypertension. Even though a supportive treatment is usually sufficient, sometimes immunosuppression is sought [180]. In a study, human umbilical vein endothelial cells were incubated with the sera from active Henoch–Schönlein purpura patients. Subsequently, nitric oxide, interleukin-8 and TNF-α increased significantly. Flavonoids were then added, which suppressed all three cytokines, suggesting their possible therapeutic role in improving microvascular inflammation [130].

The daily consumption of grape and pomegranate juices interestingly led to improvement in the endothelial function. The effects were notable as soon as four hours after juice consumption [131]. However, the study was limited by a small sample size of 30 adolescents randomized into two subgroups. Similarly, grape seed extract improved insulin concentration and insulin sensitivity in adolescents with metabolic syndrome [132].

Curcumin has been known as a potent scavenger of reactive oxygen species, which enhances the activity of antioxidants. However, its use did not have significant benefits, except for the inhibition of c-Jun N-terminal kinase, the primary kinase involved in myocardial ischemic apoptosis and caspase-3 in cardiomyocytes in patients with a tetralogy of Fallot undergoing corrective surgery [133]. Similarly, carnosine is a natural radical oxygen species scavenger. Its supplementation in patients with diabetic nephropathy resulted in a significant improvement of oxidative stress, glycemic control and renal function [134].

Salvia miltiorrhiza is an herb extract containing phenolic compounds, preventing the postoperative increase of endothelin-1. In children with congenital heart defects and pulmonary hypertension, antioxidant therapy with the herb extract reduced the myocardial damage and attenuated the postoperative vasoactive mediator imbalance [135]. Similarly, the efficacy of the plant-based bioequivalent nitrate complex, consisting of vitamins, natural antioxidants and phytophenol-rich food extracts to elevate the nitric oxide bioavailability, was studied, and an improved endothelial function was found, as well as both reduced systolic and diastolic blood pressure in hypertensive patients with daily supplementation [136].

### 6.5. Alpha-Lipoic Acid

α-lipoic acid has been identified as a powerful antioxidant by its ability to quench reactive oxygen species, decrease oxidative stress, recycle other antioxidants in the body, including vitamins C and E and glutathione, and protect against protein and lipid oxidation. It seems particularly useful in type 1 diabetes mellitus patients, where its supplementation significantly increased the glutathione levels and significantly decreased malondialdehyde, nitric oxide, tumor necrosis factor-alpha, Fas ligand, matrix metalloproteinase 2, troponin I and tumor growth factor expression. Additionally, it improved the left ventricular global peak systolic strain in diabetic patients [137]. Additionally, an antioxidant diet when supplemented with alpha-lipoic acid significantly improved the endothelial dysfunction in children with type 1 diabetes mellitus. A significant reduction in bolus insulin use was also observed [138].

## 7. Other Influences on Systemic Inflammation and Oxidative Stress in Childhood and Adolescence

Children conceived by assisted reproductive technology display vascular dysfunction. Its underlying mechanism, potential reversibility and long-term consequences for cardiovascular risk are unknown. Antioxidant administration to these children improved the nitric oxide bioavailability and vascular responsiveness in the systemic and pulmonary circulation, suggesting a relation to the decreased vascular endothelial nitric oxide synthase expression and nitric oxide synthesis [139].

Breastfeeding is commonly known as the best feeding way in the first few months of life. However, it is not always possible, and formula feeding takes place. In formula-fed infants, some studies demonstrated increased circulating inflammatory markers, such as serum monocyte chemoattractant protein-1 and uric acid, as compared to breastfed infants. Additionally, in breastfed infants, serum monocyte chemoattractant protein-1 and uric acid negatively correlated with the duration of breastfeeding [140].

Human studies have shown the beneficial effects of probiotic microorganisms on the parameters of metabolic syndrome and other cardiovascular risks. A daily intake of fermented milk with *Bifidobacterium lactis* resulted in a significant reduction in the body mass index, total cholesterol and low-density lipoprotein. Furthermore, a significant decrease in TNF-α and IL-6 was noted, indicating a potential effect of probiotics in lowering the cardiovascular risk [141]. On the contrary, another probiotic, *Lactobacillus salivarius*, failed to induce any of the desired outcomes [142].

Periodontal disease and associated inflammation have been studied in children with congenital heart defects who underwent periodontal treatment. The latter resulted in improvement of the lipid profile and inflammatory markers, indicating the importance of oral hygiene in this group of children at risk [143]. Similarly, children with obesity more commonly have chronic periodontitis in adulthood. In youth, periodontal pockets were associated with higher gingival inflammation; higher diastolic blood pressure and higher levels of IL-6, leptin, macrophage chemoattractant protein-1 and the thyroid-stimulating hormone [144].

Oxidative stress also plays an important role in acute pediatric diseases with cardiovascular implications. With the appearance of the SARS-CoV-2 virus and its epidemic, another entity in children appeared called multisystemic inflammatory syndrome (MIS-C). A child with MIS-C usually presents with fever, hypotension, severe abdominal pain and cardiac dysfunction and was positive for SARS-CoV-2 infection a few weeks before the onset of the disease. The emerging data from patients have suggested unique characteristics in the immunological response and, also, clinical similarities with other inflammatory syndromes, which can be a support as a reference in the search for the molecular mechanisms involved in MIS-C. Oxidative stress may play a very important role in the pathophysiology of MIS-C, such as oxidative stress in Kawasaki disease [10], and has recently been reviewed [11].

Finally, in young adults, higher education was associated with a favorable cardiovascular risk with a slower progression of carotid atherosclerosis [181]. Additionally, the childhood home conditions and parents’ occupations seem to have an effect on chronic inflammation and endothelial activation, with increased biomarkers of ill health when these children grow into adulthood [182].

## 8. Conclusions and Future Perspectives

Oxidative stress and systemic inflammation play an important role in the pathogenesis and disease progression of cardiovascular disease, with evidence of the deleterious effects on vascular dysfunction and atherosclerosis already at an early age. Before birth, intrauterine exposure to low-grade systemic inflammation may pose a risk for precocious atherosclerosis and impaired metabolic homeostasis. In malnourished fetuses, impaired nitric oxide production or activity seems to be a cornerstone for appropriate endothelial function, lost during mitochondrial damage. Preterm infants seem to be particularly susceptible to the damaging effects of oxidative stress, caused by hypoxic states or hyperoxia exposure. The consequences of altered fetal programming due to an unfavorable prenatal environment were also demonstrated later in life and are a promising topic for studies on antioxidant use.

In the pathogenesis of hypertension, excessive oxygen-free radical production due to the imbalance between prooxidation and antioxidation, as well as intensive lipid peroxidation, seem to contribute an important role. The state of oxidative stress correlated with organ damage, including left ventricular hypertrophy and carotid intima-media thickness in hypertensive children. Similarly, noninvasive urinary biomarkers of oxidative stress were correlated to an arteriosclerosis index. Antioxidant mechanisms, such as thiol/disulphide homeostasis and glutathione homeostasis, also play a role in pediatric hypertension.

Obesity, diabetes mellitus types 1 and 2 and the metabolic syndrome were linked to low-grade systemic inflammation, endothelial activation and a state of oxidative stress. The adverse cardiovascular effects are modulated by several biomarkers, such as adipokines, oxidized low-density lipoproteins, chemerin, catestatin, homocysteine, soluble E-selectin, IL-6 and tumor necrosis factor alpha, amongst others. Equally, endogenous antioxidants such as nitric oxide, arginine-derived polyamines, carotenes and tocopherols, vitamin D and bilirubin were altered in the cardiometabolic cluster of diseases. Studies on antioxidant use in these patients are often scarce and/or contradicting. For example, L-arginine supplementation reversed endothelial dysfunction and attenuated high blood pressure in hypertensive children after cardiac transplantation, yet did not improve the endothelial function in children with chronic renal failure.

Children with chronic kidney disease and those undergoing renal replacement therapy are important to consider, as they demonstrated increased levels of oxidative stress and systemic inflammation due to the disease and due to therapy. For instance, hemodialysis was linked to an increased loss of antioxidants and antioxidative vitamins, resulting in episodes of oxidative stress, which correlated with the degree of cardiac dysfunction. Immunosuppressive therapy with tacrolimus or cyclosporine was similarly associated with a hampered nitric oxide availability and increase in reactive oxygen species, contributing to posttransplant hypertension.

Exercise and diet interventional programs have been shown to counteract many of the adverse cardiovascular processes mentioned above, including systemic low-grade inflammation and oxidative stress. Exercise enhances the total antioxidant capacity and also leads to reduced reactive oxygen species generation at rest. In addition, it may diminish the necessary levels of antioxidants to achieve redox homeostasis at rest. A “dietary approach to stop hypertension” diet has been linked to a reduction of systemic inflammation among adolescents with metabolic syndrome. Additionally, a short-term diet and exercise intervention led to significant reductions in serum oxidative stress and endothelial activation indicators.

Despite a notable contribution of oxidative stress and systemic inflammation in many pediatric diseases, only a few studies have reported positive results from antioxidant use. Two reviews have discussed oxidative stress as a therapeutic target in cardiovascular diseases, with an emphasis on diet, nutraceutical and novel therapeutic strategies using micro-RNAs and nanoparticles [183,184]. Cardiac oxidative stress has been reviewed and described the benefits of creatine supplementation, omega-3 fatty acids, micro-RNAs and antioxidant supplementation, in addition to physical exercise [185]. Antioxidants that enhance endothelium-derived nitric oxide bioavailability have been previously reviewed, and the studies strongly suggest a beneficial effect of vitamin C and α-tocopherol, the principal component of vitamin E [186].

Vitamin C provided some benefits in a subgroup of patients after the Fontan procedure and blocked the acute hyperglycemic impairment of endothelial function in adolescents with type 1 diabetes mellitus. Vitamin E improved the therapeutic effect of recombinant human erythropoietin in children with chronic renal failure on hemodialysis. Antioxidant vitamins C and E supplementation for 6 weeks, in addition to a low-fat diet, improved the endothelial function in children with hyperlipidemia [187]. Zinc supplementation contributed to the improved levels of adipokines and some biomarkers associated with oxidative stress. L-arginine supplementation in hypertensive patients after a cardiac transplantation reversed the endothelial dysfunction and attenuated high blood pressure. The daily consumption of grape and pomegranate juices in adolescents led to improvement in the endothelial function. Dietary supplementation with carnosine, a natural radical oxygen species scavenger in patients with diabetic nephropathy, resulted in improvements of oxidative stress, glycemic control and renal function.

Antioxidant therapy with the Salvia miltiorrhiza herb extract reduced the myocardial damage and attenuated to a postoperative vasoactive mediator imbalance in children with congenital heart defects and pulmonary hypertension. Daily supplementation with a plant-based bioequivalent nitrate complex consisting of vitamins, natural antioxidants and phytophenol-rich food extracts improved the endothelial function, as well as reduced the blood pressure in hypertensive patients. An antioxidant diet, supplemented with alpha-lipoic acid, improved the endothelial dysfunction in children with type 1 diabetes mellitus. Antioxidant administration to children conceived by assisted reproductive technology improved the nitric oxide bioavailability and vascular responsiveness in the systemic and pulmonary circulations. The daily intake of fermented milk with *Bifidobacterium lactis* resulted in significant decreases in TNF-α and IL-6, indicating a potential effect of probiotics in lowering the cardiovascular risk. However, a supplementation with *Lactobacillus salivarius* did not result in similar outcomes.

In conclusion, systemic inflammation and oxidative stress are involved in the pathogenesis and cardiovascular manifestations of several chronic childhood diseases. The introduction of methods of oxidant and antioxidant status determination into clinical practice gives future perspectives to better evaluate low-grade inflammation in chronic disease and, also, to provide novel research on antioxidant uses in pediatric cardiovascular diseases.

## Figures and Tables

**Figure 1 antioxidants-11-00894-f001:**
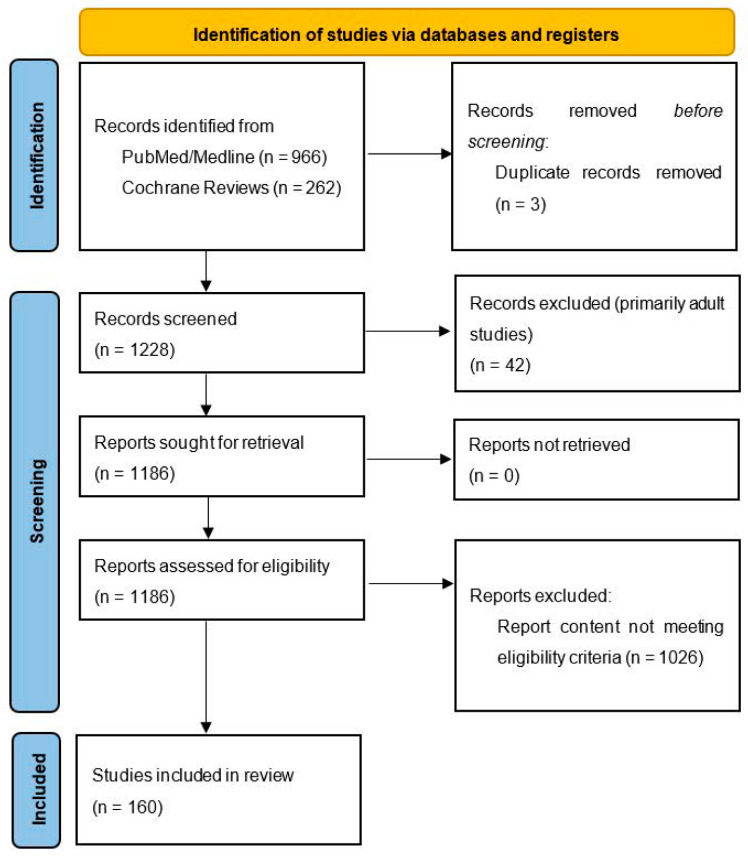
Prisma 2020 flow diagram. From Page, M.J.; Mckenzie, J.E.; Bossuyt, P.M.; Boutron, I.; Hoffmann, T.C.; Mulrow et al. The PRISMA 2020 statement: an updated guideline for reporting systematic reviews. Systematic Reviews 2021, 10 [165].

**Table 1 antioxidants-11-00894-t001:** Table of reported studies, reported by population, comparisons and study outcomes. Numbers as reported in the manuscript.

Author, Year	Country	Population	Comparison	Outcomes
Original Research				
Hashemi, 2021 [3]	Iran	108 children (51 ♂), aged 6 to 18 years	Urinary phthalate metabolites, cardiometabolic risk factors and oxidative stress markers	Exposure to phthalates correlates with cardiometabolic risk and oxidative stress markers (superoxide dismutase, malondialdehyde).
Eikendal, 2015 [4]	Netherlands	139 healthy children (59 ♂), 8 years old	Circulating chemokines and vascular characteristics	The chemokine RANTES contributes to pre-atherosclerotic inflammatory vascular changes in youth
Kelishadi, 2007 [5]	Iran and Canada	512 children (254 ♂), aged 10 to 18 years	C-reactive protein, oxidative stress markers and atherosclerotic risk factors	Oxidative stress and CRP may interact in the early inflammatory process of atherosclerosis
Suano de Souza, 2013 [7]	Brazil	35 (15 ♂) children with elevated and 29 (14 ♂) with normal homocysteine levels, mean age 8.6 years	Homocysteine, oxidative stress, carotid IMT and endothelial reactivity	No differences observed in vascular parameters and homocysteine levels in prepubertal children
Wilders-Truschnig, 2008 [8]	Austria and Luxemburg	30 obese (mean age 12.8 years) and 30 normal weight children (mean age 14.4 years)	IgG antibodies against food antigens, CRP and carotid IMT	Obese children have higher IgG antibody values against food antigens, associated with systemic inflammation and carotid IMT
Mendez-Cruz, 2007 [12]	Mexico	Human umbilical vein endothelial cells (HUVEC) of 11 mothers with strong family history of myocardial infarction	Expression of CD40 and CD40 ligand, CD80, CXCL8, tissue factor, and mono-/lymphocyte adhesion to stimulated HUVEC	HUVEC from newborns with a strong family history of MI show basal proinflammatory state and respond to proatherogenic stimuli
Schoeps, 2019 [13]	Brazil	44 (19 ♂) preterm children and 30 (13 ♂) term children, aged 5 to 9 years	Evaluation of CVD risk markers in children born term and preterm	Prepuberal preterm children show high myeloperoxidase concentrations, associated with inflammation and oxidative stress
Sakka, 2010 [14]	Greece	106 children born via IVF (48 ♂) and 68 matched controls (33 ♂), aged 4 to 14 years	IVF and insulin resistance, systemic inflammation, MetS	Children born via IVF display increased BP, but not insulin resistance or low-grade systemic inflammation
Kelishadi, 2009 [15]	Iran	112 children of parents with premature coronary heart disease, aged 12 to 18 years and 127 matched controls	Family history of premature coronary heart disease and inflammation, oxidation, echocardiography	Clusters of inflammatory factors and markers of oxidation, carotid IMT and left ventricular mass associate with family history of coronary heart disease
de Giorgis, 2009 [16]	Italy	24 prepubertal children with positive family history of premature CVD (10 ♂) and 25 healthy prepubertal controls (11 ♂)	Carotid IMT in children of family history of premature CVD and relationship to insulin resistance, oxidant status, lipid profile	Signs of precocious CVD risk are detectable in children with family history of premature CVD already during prepuberty
Craig, 2018 [17]	South Africa	55 boys with maternal and/or lifestyle risk for CVD, aged 6 to 8 years and 26 boys without maternal risk	Associations between oxidative stress and vascular functions in boys, stratified by maternal risk	Oxidative stress is and early mediator of vascular changes in the studied population
Gonzalez-Enriquez, 2008 [18]	Mexico	62 obese (34 ♂) and 21 lean controls (12 ♂), aged 6 to 19 years	Relationship between carotid IMT and functional polymorphic variants in genes for chemokines and proinflammatory cytokines associated with CVD risk	Genetic markers of an increased inflammatory response are already present in obese children and adolescents
Coelho, 2006 [19]	Portugal	49 healthy adolescents	Genetic polymorphisms, somatic characteristics, blood pressure, biochemical markers of oxidative stress and CVD risk	Carriers of angiotensin-converting enzyme DD and haptoglobin 2-2 genotypes have a higher pro-oxidant status
Guerra, 2000 [20]	Portugal	51 children, aged 9 to 12 years	Deleterious effect of reactive oxygen species on lipids, arterial endothelium and CVD	Polymorphic genetic variants of low molecular acid phosphatase and protein phosphotyrosine phosphatase are associated with oxidative stress indicators
Oztezcan, 2002 [21]	Turkey	NA	Lipid peroxidation and antioxidant system in children of hypertensive and normotensive parents	Serum malondialdehyde levels are increased in children of hypertensive parents
Hapyn, 2000 [22]	Poland	56 children with family history of hypercholesterolemia or early atherosclerosis and 20 children without positive family history	Intensity of lipid peroxidation and the activity of antioxidant enzymes in children with risk of early atherosclerosis	Children with positive family history of hypercholesterolemia and early atherosclerosis may demonstrate intensive lipid peroxidation
Kolesnikova, 2014 [23]	Russia	185 adolescents with essential hypertension (125 ♂) and 60 control subjects (36 ♂), aged 14 to 17 years	Emotional status, BP and lipid peroxidation-antioxidant protection system in adolescents	Opposite correlations between the emotional status parameters and components of lipid peroxidation–antioxidant protection system exist
Ostalska-Nowicka, 2021 [24]	Poland	65 children with essential hypertension (32 ♂) and 44 normotensive controls (20 ♂), aged 6 to 18 years	Dental caries-related primary hypertension in children and adolescents	Tooth decay in children and adolescents may be a trigger factor of essential hypertension
Sladowska-Kozlowska, 2015 [25]	Poland	126 children with arterial hypertension (95 ♂) and 83 healthy controls (40 ♂), aged 5 to 18 years	Endothelial nitric oxide (eNOS) gene polymorphism G894T and 24-h ambulatory blood pressure, carotid IMT, left ventricular mass, oxidative stress and urinary albumin secretion	The eNOS T allele is not more prevalent among hypertensive children than among healthy ones, but it is associated with early vascular damage in children with primary hypertension
Barath, 2006 [26]	Hungary	49 adolescents with essential hypertension (35 ♂, mean age 14.0 years), 79 with obesity-induced hypertension (56 ♂, mean age 14.0 years), 20 uremic patients (11 ♂ mean age 14.9 years), 60 obese patients (33 ♂, mean age 13.2 years and 57 age-matched controls (33 ♂, mean age 13.7 years)	Roles of oxidative stress and paraoxonase 1 in uremic, essential or obesity-induced hypertension	No significant correlation between the biochemical parameters and neither genotypes nor paraoxonase activities
Litwin, 2010 [27]	Poland	44 children with untreated primary hypertension (36 ♂, mean age 13.7 years) and 30 healthy controls (13 ♂, mean age 12.7 years)	Inflammatory activation in children with primary hypertension	Lack of correlation between CRP and chemokines suggests that vascular inflammation in primary hypertension precedes the systemic inflammatory changes
Turi, 2003 [28]	Hungary, USA	52 children with essential hypertension (37 ♂, mean age 14.4 years) and 48 controls (27 ♂, mean age 14.3 years)	Plasma nitrites and nitrates, lipid peroxidation end-products and redox status of red blood cell glutathione in juvenile essential hypertension	Presence of systemic oxidative stress in hypertensive children and adolescents, irrespective of their body mass index
Sladowska-Kozlowska, 2012 [29]	Poland	86 children with primary hypertension (66 ♂), aged 5 to 17 years	Primary hypertension, target organ damage and oxidative stress (reduced glutathione, glutathione peroxidase activity, thiobarbituric acid reactive substances, asymmetric dimethyloarginine and oxLDL) before and after 12 months of (non)pharmacological therapy	Oxidative stress in children with primary hypertension correlates with target organ damage, metabolic abnormalities, changes in fat amount and improvement of insulin sensitivity, but not with BP decrease
Yamano, 2015 [30]	Japan	85 children (45 ♂), mean age 6.9 years	Arteriosclerosis index and oxidative stress markers in school children	Urinary 8-isoprostaglandin F2α may serve as early marker in predicting risk in children of developing lifestyle-related diseases
Cakici, 2018 [31]	Turkey	30 children with primary hypertension (19 ♂, mean age 14.8 years) and 30 healthy controls (17 ♂, mean age 14.3 years)	Level of dynamic thiol/disulphide homeostasis in adolescents with newly diagnosed primary hypertension	Shift towards disulphide formation in adolescent patients with primary hypertension
Stoppa-Vaucher, 2012 [32]	Switzerland	32 obese children (19 ♂, mean age 11.7 years) and 29 lean controls (19 ♂, mean age 11.3 years)	Inflammatory and prothrombotic states in obese children of European descent	Obese children present with inflammatory and prothrombotic states as early as 6 years of age
Valle Jimenez, 2007 [33]	Spain	46 obese prepubertal children (19 ♂) and 46 lean controls (19 ♂), aged 6 to 9 years	Relationship between endothelial dysfunction and both insulin resistance and inflammation in prepubertal obese children	An association between endothelial dysfunction, insulin resistance, inflammation and inappropriate fibrinolysis was established in the children studied
Mauras, 2010 [34]	USA	115 obese (59 ♂) and 88 lean children (47 ♂), aged 7 to 18 years	Markers of inflammation and prothrombosis in obese children without established metabolic syndrome comorbidities	Childhood obesity is associated with a proinflammatory and prothrombotic state before other comorbidities of the MetS are present and even before the onset of puberty
Morandi, 2020 [35]	Italy	152 obese children and adolescents (79 ♂, mean age 11.9 years)	Association between total antioxidant capacity and BP	The systemic anti-oxidant capacity is inversely associated with systolic BP and pulse pressure in children and adolescents with obesity
Ovunc Hacihamdioglu, 2015 [36]	Turkey	24 hypertensive obese children (16 ♂, mean age 13.1 years), 22 normotensive obese (14 ♂, mean age 11.5 years) and 27 healthy children (13 ♂, mean age 11.2 years)	Effect of obesity and anti-hypertensive treatment on urinary Th1 chemokines	Th1 cells could be activated in obese hypertensive children before the onset of clinical indicators of target organ damage
Syrenicz, 2006 [37]	Poland	281 obese children (151 ♂), aged 6 to 18 years	Role of low-grade, systemic inflammation and endothelial activation in the modulation of BP independently of other traditional risk factors	Low-grade inflammation may play a role in the modulation of arterial BP relatively early in life
Aburawi, 2019 [38]	United Arab Emirates	967 thin, normal, overweight and obese children, aged 7 to 16 years	Biomarkers for glycemic control, lipid metabolism, systemic inflammation, endothelial dysfunction and hepatic cholestasis	Children with excess fat had increased risks for developing systemic inflammation, dyslipidemia, endothelial dysfunction, cholestasis and diabetes
Al-Shorman, 2017 [39]	Jordan	29 lean (17 ♂), 29 obese (17 ♂) and 29 severely obese children (17 ♂), aged 10 to 15 years	Levels of carotid IMT, endothelial dysfunction and inflammatory biomarkers	Biomarkers of inflammation and endothelial dysfunction were higher in obese schoolchildren, but are not increased by the degree of obesity nor the MetS cluster
Giannini, 2008 [40]	Italy	53 obese pre-pubertal children (27 ♂, mean age 8 years) and 41 healthy pre-pubertal controls (21 ♂, mean age 7 years)	Relationship between carotid IMT, insulin resistance and oxidant status	Early changes in glucose metabolism and an alteration of oxidant–antioxidant status may be present in obese pre-pubertal children, which could lead to increased carotid IMT and early CVD
Sinaiko, 2005 [41]	USA, Sweden, Norway	Cohort of 295 adolescents (169 ♂, mean age 15 years)	Relation of fatness and insulin resistance and their interaction with CVD risk factors, inflammatory factors and oxidative stress	Insulin resistance may be acting interactively with fatness
Jung, 2009 [42]	Sweden	79 ♂ adolescents, aged 13 to 17 years	Anthropometric measures, adiponectin, stromal-derived factor (SDF-1) and soluble E-selectin as parameters for beginning of insulin resistance and endothelial damage	SDF-1 might be a new marker for diagnosis of obesity-related diseases and help understand pathophysiologic mechanisms
Codoner-Franch, 2012 [43]	Spain	54 severely obese (33 ♂) and 44 healthy children (25 ♂), aged 7 to 14 years	Plasma level of advanced oxidation protein products (mAOPPs) and relation to metabolic risk factors	Determination of plasma mAOPPs levels is an easy method that can evaluate protein oxidation and provide information related to metabolic risk and treatment effectiveness
Okur, 2013 [44]	Turkey	27 prepubertal obese children (59% ♂, mean age 7 years,) and 30 healthy children (55% ♂, mean age 7 years)	Circulating oxidized low-density lipoprotein (LDL) concentrations and the carotid IMT and possible association with carotid atherosclerosis	Oxidation of LDL starts early in obese children, but the carotid IMT is not significantly affected
Elmas, 2017 [45]	Turkey	65 children with exogenous obesity (34 ♂) and 64 healthy children (27 ♂), aged 5 to 17 years	Thiol/disulphide homeostasis as a novel and sensitive marker of oxidative stress and its relationship with some inflammatory and cardiovascular markers	The impairment in thiol/disulphide homeostasis, which is indicative of oxidative stress, is associated with inflammation in obesity
Correia-Costa, 2016 [46]	Portugal, Germany	89 overweight (43 ♂), 61 obese (40 ♂) and 163 lean children (83 ♂), aged 8 to 9 years	Oxidative stress and NO production/ metabolism and correlations with cardiometabolic risk and renal function	Oxidant status and NO are increased in relation to fat accumulation and translate into higher values of cardiometabolic risk markers and affect renal function
Codoner-Franch, 2011 [47]	Spain	60 obese and 42 lean children, aged 7 to 14 years	Childhood obesity and modification of blood polyamines, circulating markers of oxidative and nitrosative stress and endothelial dysfunction	Polyamine levels are increased in childhood obesity and correlate to markers of oxidative/nitrosative stress and angiogenesis
Landgraf, 2012 [48]	Germany	69 lean and 105 obese children, aged 7 to 18 years	Association of chemerin with obesity and early-onset metabolic and vascular sequelae	There is an association of chemerin with obesity and inflammatory and endothelial activation markers
Niklowitz, 2018 [49]	Germany	88 overweight children (39 ♂, mean age 11.9 years) and 23 lean children (10 ♂, mean age 11 years)	Relationships between chemerin, parameters of fat mass and MetS in obese children before and after weight reduction	Chemerin is related to parameters of central fat mass and MetS
Wojcik, 2020 [50]	Poland	23 obese children (10 ♂, mean age 9 years)	Circulating chemerin level and 24 h blood pressure monitoring results and IMT	Elevated chemerin level may be associated with increased systolic BP in obese children
Simunovic, 2019 [51]	Croatia	92 obese children (52 ♂) and 39 healthy, lean controls (18 ♂), aged 10 to 18 years	Catestatin in pediatric obesity, metabolic syndrome and correlations between catestatin and CVD risk	Serum catestatin concentrations are decreased in obese children and adolescents
Makni, 2013 [52]	France	60 obese children (24 ♂), 54 obese children with MetS (33 ♂) and 37 lean controls (19 ♂), mean age 13 years	Correlation of resistin with inflammatory and cardiometabolic markers in obese adolescents with and without MetS	Plasma resistin showed higher correlations with anthropometric parameters, lipid profiles, systolic and diastolic BP and pro-inflammatory cytokines in obese children with MetS
Stringer, 2009 [53]	Canada	24 children with T2DM, 19 obese children and 34 lean controls, aged 12 to 15 years	CVD risk in children with and without T2DM or obesity by comparing pro- and anti-inflammatory adipokines, markers of oxidative stress and the plasma phospholipid fatty acid profile	Altered plasma adipokines and markers of oxidative stress suggest increased risk of CVD in youth with obesity or T2DM
Stelzer, 2012 [54]	Austria, USA	256 overweight and obese children (114 ♂), aged 8 to 18 years and 67 lean controls	Adipocytokines, interleukin-6 and leptin in overweight/obese and normal-weight subjects	Interleukin-6 is increased with the grade of overweight in every age group. Leptin is essentially involved in the early priming phase of obesity-related inflammation
Mohanraj, 2013 [55]	Korea, USA	41 overweight (19 ♂), 56 obese children (25 ♂) and 100 lean controls (49 ♂), mean age 12 years	Comparison of obese adolescents to normal counterparts for total insulin-like growth factor-binding protein 3 (IGFBP-3) levels and proteolyzed IGFBP-3 in circulation	IGFBP-3 inhibits cytokine-induced insulin resistance and early manifestations of atherosclerosis
Ostrow, 2011 [56]	USA	42 obese children (25 ♂, mean age 12.8 years) and 34 non-obese children (22 ♂, mean age 11.8 years)	Oxidative stress in obese vs. non-obese children with correlation to adiposity, obesity-related metabolic abnormalities and ambulatory BP	There is a correlation between oxidative stress, adiposity and BP in children
Selvaraju, 2019 [57]	USA	24 overweight (14 ♂), 23 obese (9 ♂) and 41 normal weight children (23 ♂), mean age 8 years	Urinary biomarkers of inflammation (CRP, interleukin-6, and α-1-acid glycoprotein), and oxidative stress markers (8-isoprostane, 8-hydroxy-2′-deoxiguanosine and endothelin-1)	Urinary biomarkers of inflammation and oxidative stress are elevated in obese children and correlate with a marker of endothelial dysfunction
Metzig, 2011 [58]	USA	34 obese children (17 ♂, mean age 12.4 years)	Endothelial function, glucose, insulin, CRP, interleukin-6, circulating oxidized low-density lipoprotein and myeloperoxidase in a fasting state and at 1- and 2-h following glucose ingestion	An acute oral glucose load does not reduce endothelial function or increase levels of inflammation or oxidative stress in obese youth
Codoner-Franch, 2012 [59]	Spain	66 obese children (41 ♂) and 39 normal weight children (19 ♂), aged 8 to 13 years	Assessment of CVD risk in severely obese children	Insufficient 25(OH)D levels were detected in severely obese children with increased markers of oxidative/nitrosative stress, inflammation and endothelial activation.
Rajakumar, 2020 [60]	USA	225 overweight or obese children, aged 10 to 18 years	Effect of vitamin D3 supplementation on vascular and metabolic health of vitamin D–deficient	Correction of vitamin D deficiency in overweight and obese children did not affect measures of arterial endothelial function or stiffness, systemic inflammation, or lipid profile, but lowered BP and fasting glucose concentration improved insulin sensitivity
Al-Daghri, 2016 [61]	Saudi Arabia	224 children and 140 adults	Association of vitamin B12 with pro-inflammatory cytokines and biochemical markers related to cardiometabolic risk	Maintaining adequate vitamin B12 concentrations may lower inflammation-induced cardiometabolic risk
Economou, 2004 [62]	Greece	72 prepubertal obese children (33 ♂), aged 7 to 10 years and 42 controls	Obese compared with lean children for a possible relation among plasma total homocysteine, monocyte chemoattractant protein-1 (MCP-1) and RANTES	Negative association between circulating total homocysteine and proinflammatory chemokines MCP-1 and RANTES in prepubertal lean, but not in obese children
Giannini, 2009 [63]	Italy	40 obese children (19 ♂, mean age 8.5 years), 40 lean children (22 ♂, mean age 8.7 years) and 40 matched controls (24 ♂, mean age 8.1 years)	Oxidant–antioxidant status, inflammatory markers and carotid intima-media thickness	Prepubertal lean and obese children present increased oxidative stress and impaired inflammation and insulin sensitivity
Maggio, 2018 [64]	Switzerland, Italy	48 children (28 ♂, mean age 9.7 years), 35 after behavioral obesity intervention and 13 controls	Cytokines, adiponectin, neutrophil product MMP-8, carotid IMT, flow-mediated dilation, nitroglycerin-mediated dilation, arterial stiffness, pulse wave velocity, resting and 24-hour BP after a 6-month behavioral intervention to treat obesity	Behavioral interventions resulted in a paradoxical increase in some biomarkers in children, with potentially beneficial effects detected with CCL2 changes
Marti, 2021 [65]	Spain	29 children in a lifestyle intervention (14 ♂), aged 7 to 16 years	Lifestyle intervention with a 2-month intensive phase and a subsequent 10-month follow-up with a moderate calorie-restricted diet, physical activity and nutritional education	Higher lipopolysaccharide binding protein and chemerin concentrations were associated with MetS in children with abdominal obesity during a lifestyle intervention
Akinci, 2007 [66]	Turkey	50 children (22 ♂, mean age 11.5 years) of parents with MetS and 38 matched controls	Comparison of anthropometric measurements, BP, echocardiography, flow-mediated vasodilatation and metabolic fasting blood measurements	Children of patients with MetS have higher values of the serum markers of inflammation
Olza, 2015 [67]	Spain	146 overweight (65♂), 295 obese (161 ♂) and 236 normal weight children (127♂), aged 4 to 12 years	MetS score traits, markers of inflammation, endothelial damage and CVD risk	MetS score with specific risk biomarkers of inflammation, endothelial damage and CVD are useful in the early identification of children at increased risk of metabolic dysfunction
Gonzalez-Jimenez, 2016 [68]	Spain	976 adolescents, mean age 13.2 years; 930 in non-MetS group (441 ♂) and 46 in MetS group (16 ♂)	Measurement of interleukin-6 (IL-6), tumor necrosis factor-a (TNFa), high- sensitivity CRP and ceruloplasmin	Subjects with MetS exhibited higher levels of TNF2a, IL-6, CRP and ceruloplasmin
Kelly, 2006 [69]	USA	11 normal weight and healthy children (3 ♂), 13 overweight and healthy (7 ♂) and 10 overweight with the MetS (overweight + MetS) (5 ♂), aged 8 to 14 years	Body composition, BP, lipids, glucose tolerance, markers of insulin resistance, oxidative stress, and adipokines	Oxidative stress and adverse adipokine profile characterize the MetS in children
Scuteri, 2011 [70]	USA, Italy	6148 children and adults, aged 14–102 years, enrolled in the SardiNIA Study	MetS components, common carotid artery diameter, IMT and aortic pulse wave velocity, adiponectin, leptin, high-sensitivity CRP, monocyte chemoattractant protein 1 and interleukin-6 levels	MetS and specific cytokine patterns are associated with arterial aging, the increases in arterial stiffness and thickness
Lin, 2009 [71]	Taiwan	4723 children, aged 12 to 17 years	Measurements of various serum hepatic profiles and metabolic risks from Health and Nutrition Examination Survey 1999–2004	Serum bilirubin is inversely associated with insulin resistance and MetS among children and adolescents
Ruperez, 2020 [72]	Spain	1444 children (706 ♂), aged 3 to 17 years	Plasma antioxidants, oxidative stress biomarkers and associations with pro-inflammatory and endothelial damage biomarkers	Antioxidants and oxidative stress biomarkers showed novel associations with several pro-inflammatory and endothelial damage biomarkers
Huerta-Delgado, 2020 [73]	Mexico	21 children with T2DM (10 ♂), 19 with MetS (11 ♂) and 17 healthy controls (8 ♂), aged 6 to 16 years	Associations among circulating irisin levels, soluble cell adhesion molecules (sCAMs) and inflammatory cytokines	MetS and T2DM patients have lower serum irisin levels. T2DM subjects have lower concentrations of most sCAMs compared to MetS patients
Reilly, 1998 [74]	USA, Italy, Australia	38 children and adults (24 ♂), aged 3 to 24 years with homozygous familial hypercholesterolemia and 24 adults (16 ♂) with moderate hypercholesterolemia	Comparison of hyper- to normocholesterolemic control subjects for F2 isoprostanes iPF2a-III, iPF2a-VI and arachidonic acid (AA)	Asymptomatic patients with moderate and severe hypercholesterolemia have evidence of oxidant stress in vivo
Charakida, 2009 [75]	Greece	38 children with familial hypercholesterolemia (19 ♂, mean age 14.8 years) and 41 healthy controls (22 ♂, mean age 15.4 years)	Endothelium dependent reactive hyperemia, endothelium-independent nitrate hyperemia dilatation, inflammatory and hemostatic parameters	Inflammatory and thrombotic processes are associated with vascular dysfunction in children with familial hypercholesterolemia
Holven, 2006 [76]	Norway	33 children with familial hypercholesterolemia (13 ♂, mean age 14 years), 14 hypercholesterolemic adults (7 ♂, mean age 47 years) and 30 controls	Gene expression of chemokines in peripheral blood mononuclear cells from clinically healthy children with and without heterozygous familial hypercholesterolemia	A role of inflammation in the early stages of atherogenesis, possibly involving monocyte-derived RANTES
Loffredo, 2012 [77]	Italy	20 children with hypercholesterolemia (10 ♂, mean age 10 years), 20 with obesity (10 ♂, mean age 10 years), 20 obese hypercholesterolemic children (10 ♂, mean age 10 years) and 40 healthy controls (19 ♂, mean age 10 years)	Interplay among oxidative stress, NOX2, the catalytic core of nicotinamide-adenine dinucleotide phosphate oxidase, and endothelial dysfunction in children with obesity and/or hypercholesterolemia	NOX2-generating oxidative stress may have a pathogenic role in the functional changes of the arterial wall occurring in obesity and/or hypercholesterolemia
Martino, 2008 [78]	Italy	50 children with hypercholesterolemia (19 ♂, mean age 10 years) and 50 healthy controls (25 ♂, mean age 9.2 years)	Comparison of flow-mediated dilation, IMT, lipid profile, urinary isoprostanes, markers of oxidative stress and platelet expression of gp91phox	Gp91phox-mediated oxidative stress may have a pathogenic role in the anatomic and functional changes of the arterial wall occurring in children with premature atherosclerosis
Ece, 2006 [79]	Turkey	29 children with CRF (14 ♂, mean age 10.2 years) and 25 healthy controls (14 ♂, mean age 8.3 years)	Markers of oxidative stress (superoxide dismutase, catalase activities, glutathione and malondialdehyde levels), inflammation and early cardiovascular abnormalities	Increased oxidant stress and inflammation together with early cardiovascular damage were found in children with CRF
Ece, 2006 [80]	Turkey	29 children with CRF (14 ♂, mean age 10.2 years) and 25 healthy controls (14 ♂, mean age 8.3 years)	Investigation of PON activity, total antioxidant response, total peroxide, oxidative stress index and some pro-oxidant cytokines	Low levels of serum albumin and high levels of uremic metabolites might be responsible for increased oxidative stress in children with CRF
Hamed, 2012 [81]	Egypt	40 children with CKD (25 ♂) and 20 healthy children (10 ♂), aged 6 to 15 years	Effect of CKD and hemodialysis (HD) on hypoxia and oxidative stress biomarkers	Patients with CKD succumb considerable tissue hypoxia with oxidative stress. HD ameliorates hypoxia but lowers antioxidants (decreased levels of hypoxia induced factor-1α and total antioxidant capacity)
Zachwieja, 2005 [82]	Poland	21 children with ESRD; 10 on HD (4 ♂, mean age 15.2 years), 11 on peritoneal dialysis (8 ♂, mean age 10.2 years) and 9 healthy controls (4 ♂, mean age 8.9 years)	Cytokine synthesis and oxidative stress in peripheral blood lymphocytes of ESRD	Patients with ESRD, especially those treated with HD, present increased oxidative stress in T lymphocytes, which may lead to decreased cytokine synthesis and abnormal immune response
Maciejczyk, 2018 [83]	Poland	25 children with CKD (15 ♂) and 25 healthy controls (15 ♂), aged 7 to 18 years	Evaluation of oxidative stress indicators in the non-stimulated and stimulated saliva	CKD is responsible for disturbances in salivary antioxidant systems and oxidative damage to proteins and lipids
Al-Biltagi, 2016 [84]	Egypt	30 children with ESRD on regular hemodialysis (20 ♂, mean age 13.7 years) and 30 healthy controls	Cardiac function and plasma glutathione level, a marker of oxidative stress	Significant oxidative stress was present in children with ESRD and was correlated with the degree of cardiac dysfunction
Elshamaa, 2009 [85]	Egypt	30 children on hemodialysis (17 ♂) and 20 controls (10 ♂), aged 4 to 18 years	Comparison of plasma total antioxidant capacity and lipid peroxidation products and correlation to high-sensitivity CRP	Children on HD have increased oxidative stress, which may explain the cardiovascular complications in HD patients
El-Saeed, 2015 [86]	Egypt	30 children on hemodialysis (23 ♂, mean age 8.7 years) and 30 controls	Advanced glycation end products AGEs, oxidized LDL, soluble receptor AGE as markers of oxidative stress	The low molecular weight form of AGEs is associated with oxidative stress in children receiving chronic HD
Zwolinska, 2006 [87]	Poland	32 children with moderate CRF (18 ♂, mean age 12 years), 14 children with advanced CRF (7 ♂, mean age 12 years), 21 children on hemodialysis (11 ♂, mean age 14.8 years) and 27 controls	Evaluation of the plasma, erythrocyte and dialysate levels of vitamins A and E and the plasma and dialysate levels of vitamin C as exogenous non-enzymatic antioxidants	CRF in children is associated with decreased concentrations of plasma antioxidant vitamins, particularly plasma vitamin C and erythrocyte vitamin E .
Zwolinska, 2006 [88]	Poland	10 children on continuous ambulatory peritoneal dialysis (2 ♂, mean age 13.2 years), 21 children on hemodialysis (11 ♂, mean age 14.8 years) and 27 controls	Lipid peroxidation in plasma and erythrocytes, erythrocyte antioxidant enzyme activity and concentrations of Cu and Zn and Se in erythrocytes, plasma and in dialysis fluid	Increased oxidative stress occurs in children on maintenance dialysis, independent of dialysis modality. Oxidative stress is aggravated during every single HD session in children
Badawy, 2020 [89]	Egypt	50 children with ESRD on regular hemodialysis (24 ♂), aged 5 to 16 years	Some inflammatory markers and measurement of left ventricular hypertrophy (LVH)	Elevated hsCRP and IL-18 are independent determinants of LVH in children with HD
Drozdz, 2016 [90]	Poland	65 children with stage 1 to 5 CKD (41 ♂, mean age 11.2 years)	Oxidative stress biomarkers, cardiovascular risk factors and LVH	Hypertension and dyslipidemia correlated with lipid oxidation and oxLDLs seem to be valuable markers of oxidative stress in CKD patients, correlating with LVH
Garcia-Bello, 2014 [91]	Mexico	134 children with CKD (68 ♂), aged 6–17 years	Inflammation and oxidative stress with carotid IMT (cIMT) and elasticity increment module (Einc)	cIMT and Einc strongly associate with several biochemical parameters and glutathione
Kotur-Stevuljevic, 2013 [92]	Serbia	52 children with CKD (28 ♂, mean age 14.2 years) and 36 healthy controls	Dyslipidemia and oxidative stress in the early phases of atherosclerosis	Early atherosclerosis in CKD children is caused also by dyslipidemia and oxidative stress
Cengiz, 2009 [93]	Turkey	28 patients receiving HD (15 ♂, mean age 15.1 years) and 20 healthy children (7 ♂, mean age 14.3 years)	Oxidative stress and cardiovascular risk factors	Children treated by hemodialysis are exposed to oxidative stress and chronic inflammation
Srivaths, 2010 [94]	USA	16 children receiving HD (10 ♂, mean age 17.2 years)	Inflammation, malnutrition, renal osteodystrophy and coronary calcification	Coronary calcification (CC) is prevalent in older children receiving maintenance HD with a longer dialysis vintage. Worse renal osteodystrophy control and malnutrition (low cholesterol) may contribute to CC development.
Calo, 2006 [95]	Italy, USA	8 children with renal transplant and hypertension (1 ♂, mean age 15 years) and 8 renal transplant patients without hypertension (4 ♂, mean age 16 years)	Oxidative stress signaling and posttransplant endothelial dysfunction and hypertension	Oxidative stress is associated with posttransplant hypertension in hypertensive pediatric kidney-transplant patients
Al-Mashhadi, 2018 [96]	Sweden	15 patients with unilateral congenital hydronephrosis (14 ♂) and 2 matched control groups, i.e., 8 healthy controls (6 ♂) and 8 operated controls (7 ♂), aged 1 to 12 years	Measurement of changes in arterial pressure, oxidative stress and NO homeostasis following correction of hydronephrosis	There is increased arterial pressure and oxidative stress in children with hydronephrosis
Pavlova, 2005 [97]	Bulgaria	39 patients with kidney disease (14 ♂, aged 2,6 to 17 years) and 13 healthy controls (8 ♂, aged 2 to 8 years)	Dynamics of oxidative stress in the blood and urine	Products of lipid peroxidation, intensity of chemiluminescence, total and enzyme antioxidant capacity in combination with clinical parameters are a proper test for the dynamics of oxidative stress
Biltagi, 2008 [98]	Egypt	40 children with adenoidal hypertrophy and 20 control children, aged 4 to 8 years	Obstructive sleep apnea (OSA), levels of 8-isoprostane, interleukin-6 (IL-6) and cardiac diastolic dysfunctions	OSA severity is positively correlated with the degree of elevation of 8-isoprostane and IL-6 in breath condensate of children with OSA and with the degree of cardiac dysfunction
Gozal, 2008 [99]	USA	20 children with OSA before tonsillectomy and adenoidectomy and after (12 ♂, mean age 6.5, 7.2 years, respectively) and 20 controls (12 ♂, mean age 6.4)	Interleukins -6 and -10 before and 4–6 months after tonsillectomy and adenoidectomy	Systemic inflammation is a component of OSA, even in the absence of obesity and is reversible upon treatment in most patients
Smith, 2021 [100]	USA	43 patients with OSA (16 ♂, mean age 9 years) and 53 healthy controls (22 ♂, mean age 10 years)	Structural, functional carotid changes and inflammatory profiles	IL-6 and IL-8 were associated with changes in carotid function in children with OSA
Loffredo, 2015 [101]	Italy	22 children with OSA (14 ♂, mean age 7.6 years) and 45 children with primary snoring (31 ♂, mean age 8.9 years) and 67 healthy controls (45 ♂, mean age 8.7 years)	Oxidative stress, assessed by serum isoprostanes and soluble NOX2-dp and endothelial function, assessed by flow-mediated dilation	NOX2-derived oxidative stress is involved in artery dysfunction in sleep disordered breathing children
Van Hoorenbeeck, 2012 [102]	Belgium	132 obese subjects (41 ♂, median age 15.4 years)	Sleep-disordered breathing (SDB), inflammation and oxidative stress before and after weight loss	Oxidative stress is reflected by uric acid at baseline and the concentration decreases after treatment according to improvements in SDB
Babar, 2019 [103]	USA	52 children with T1DM, aged 12 to 18 years, from these 27 with hemoglobin A1c (HbA1c) ≤8.5% and 25 with HbA1c ≥9.5%	Glycemic control, vascular oxidative stress, inflammation and vascular health	Elevated E-selectin level is an early marker of oxidative stress in T1DM patients with an elevated HbA1c level
Suys, 2007 [104]	Belgium	35 children with T1DM (16 ♂, mean age 14 years), matched with 35 non-diabetic controls (18 ♂, mean age 14 years)	Endothelial dysfunction and oxidative stress	Low Cu/Zn superoxide dismutase is a potential early marker of susceptibility to diabetic vascular disease
Sochett, 2017 [105]	Canada, UK	51 children with T1DM (25 ♂, mean age 14.8 years) and 59 healthy controls (26 ♂, mean age 13.9 years)	Systemic inflammation and vascular function	The cytokine-chemokine signature in early type 1 diabetes mellitus, prior to the development of arterial disease, is significantly different from that seen in healthy controls
Mylona-Karayanni, 2006 [106]	Greece	45 children with T1DM (27 ♂, mean age 13.8 years) and 20 healthy controls (11 ♂, mean age 13.2 years)	Oxidative stress parameters and adhesion molecules derived from endothelial/platelet activation	Decreased antioxidative protection from simultaneous nitrite and lipid hydroperoxide overproduction is evident in T1DM juveniles with a parallel endothelial/platelet activation even in the first years of the disease and contribute to the vascular complications
Stankovic, 2012 [107]	Serbia	40 children with T1DM (30 ♂, mean age 12.8 years) and 20 healthy controls (13 ♂, mean age 11.8 years)	Traditional and novel risk factors, such as anti-oxidative capacity of circulating blood and level of lipid peroxidation and atherosclerosis	Despite the documented increased oxidative stress, there was no correlation between the oxidative stress parameters and coronary IMT values
Seckin, 2006 [108]	Turkey	100 children with T1DM (54 ♂), aged 2 to 17 years	Effect of glycemic control on oxidative stress	Increased levels of MDA, ICAM-1, NO and VEGF present higher risk for atherosclerosis in poorly controlled T1DM in children
Sharma, 2013 [109]	UK	32 young patients with T1DM and 32 controls	Novel method to measure metabolites of vitamin E and its measurement	Metabolites of vitamin E are increased in children with type 1 diabetes indicating an increased oxidative stress
Chiesa, 2019 [110]	UK	70 adolescents with T1DM, aged 10 to 17 years	High-density lipoprotein and endothelial function	Impaired endothelial function in patients with high inflammatory risk score and high levels of HDL
Yau, 2017 [111]	USA	157 adolescents (112 overweight), aged 16 to 22 years	Relationships between retinal vessel diameter, physical fitness, insulin sensitivity, systemic inflammation	Improved physical fitness and insulin function reduces inflammation in adolescents and improve cerebrovascular function later in life
Roberts, 2007 [112]	USA	19 overweight children, aged 8 to 17 years	Effects of lifestyle modification on oxidative markers	Reduced several traditional and novel factors associated with atherosclerosis
Llorente-Cantarero, 2021 [113]	Spain	216 children (111 ♂), aged 6 to 14 years	Physical fitness and oxidative stress	A high physical-activity-sedentarism score is associated to a better redox profile
Kelly, 2004 [114]	USA	25 children with overweight (12 ♂, mean age 10.9 years)	Exercise in overweight children and subclinical inflammation and endothelial function	Aerobic exercise improves fitness, HDL cholesterol, endothelial function; CRP is associated with fasting insulin
Leite-Almeida, 2021 [115]	Portugal	313 children (150 overweight/obese), aged 8 to 9 years	Physical activity and nitric oxide biomarkers in non-overweight and overweight/obese	Urinary nitrates were higher in children with higher physical activity and in non-overweight
Platat, 2006 [116]	France	640 adolescents, aged 12 years	Effect of physical activity on symptoms of MetS and low-grade inflammation	In adolescents, physical activity is inversely related to HOMA and IL-6, independently of adiposity and fat localization
Liu, 2018 [117]	China	50 obese female adolescents, aged 14 to 16 years	Exercise, diet and chemerin in study group	Aerobic exercise with diet showed decreased chemerin in comparison to diet only group
Dennis, 2013 [118]	USA	112 overweight/obese children, aged 7 to 11 years	Exercise and plasma isoprostane levels	No effect of exercise on isoprostane levels; isoprostane levels were related to several cardiovascular risk at baseline
Santiprabhob, 2018 [119]	Thailand	115 obese young individuals (mean age of 60 males 12,1 years and mean age of females 12,5 years)	Lifestyle modification program and CVD markers	Healthy lifestyle program for obese youths had beneficial effects on adipocytokines, inflammatory cytokines and arterial stiffness
Cazeau, 2016 [120]	USA	2 males, 7 females with T1DM, mean age 12.9 years	Antioxidant supplementation and oxidative damage	Antioxidant therapy does not decrease oxidative damage, improve endothelial function or increase vascular repair capacity
Goldstein, 2012 [121]	USA	23 vitamin C and 21 placebo assigned subject after Fontan completion, aged 8 to 25 years	Vitamin C and function health status in Fontan-palliated patients	Shorter therapy with vitamin C does not alter endothelial function or exercise capacity; it might provide benefit to those with abnormal vascular function
Hoffman, 2012 [122]	USA	5 male, 3 female adolescents with T1DM, mean age 14.2 years	Ascorbic acid and hyperglycemia	Antioxidant treatment with vitamin C blocs acute hyperglycemic impairment of endothelial function
Chiarelli, 2004 [123]	Italy	12 adolescents with T1DM, aged 11 to 21 years	Vitamin E and early signs of microangiopathy	Early diabetic angiopathy in adolescents cannot be modified by vitamin E
Németh, 2000 [124]	Hungary	10 children, mean age 15.2 years	Chronic hemodialysis, recombinant human erythropoietin and vitamin E	Vitamin E supplementation improved the therapeutic effect of rhEPO in patients with chronic renal failure on hemodialysis
Kelishadi, 2010 [125]	Iran	60 obese children, aged 6 to 10 years	Zinc and insulin resistance, oxidative stress and inflammation in obese	Significant decrease was documented for Apo B/ApoA-I ratio, ox-LDL, leptin and malondialdehyde, total and LDL-cholesterol after receiving zinc
Soongswang, 2005 [126]	Thailand	15 patients with idiopathic dilated cardiomyopathy, aged 0.6 to 16.3 years	Coenzyme Q10, cardiac function and quality of life	CoQ10 may improve NYHA class and cardiothoracic ratio and shorten ventricular depolarization
Serag, 2021 [127]	Egypt	49 T1DM pediatric patients and 7 healthy controls	Coenzyme Q10 and various metabolic parameters	CoQ10 had no favorable effect on endothelial dysfunction or metabolic parameters
Bennett-Richards, 2002 [128]	UK	21 normotensive children with CRF, aged 7 to 17 years	L-arginine and endothelial function	Endothelial function was not improved with L-arginine
Lim, 2004 [129]	USA	8 children after heart transplantation (4 ♂), aged 9 to 29 years and 15 healthy controls	L-arginine, endothelial function and high BP	Oral L-arginine therapy reverses endothelial dysfunction and attenuates high blood pressure in hypertensive cardiac transplant recipients
Bo, 2012 [130]	China	10 children with typical nonthrombocytopenic purpura and 10 healthy controls	Supernatant IL-8, TNF-alpha and NO levels of human umbilical vein endothelial cells and sera of participants and extracted flavonoids from *B. bipinnata*	Extracted flavonoids from *B. bipinnata* improve microvascular inflammation in Henoch–Schönlein patients
Hashemi, 2010 [131]	Iran	30 adolescents with MetS, aged 12 to 15 years	Grape and pomegranate juice and endothelial function	Daily consumption of diets rich in antioxidants might improve endothelial function in adolescents with MetS
Mohammad, 2021 [132]	Iran	42 adolescents with MetS, aged 13 to 19 years	Grape seed extract effect on insulin resistance	Grape seed extract improves insulin concentration and insulin resistance in adolescents with metabolic syndrome
Sukardi, 2016 [133]	Indonesia	45 children after tetralogy of Fallot corrective surgery, aged 1 to 6 years	Curcumin and the concentrations of malondialdehyde and glutathione, activity of nuclear factor-kappa B, c-Jun N-terminal kinase, caspase-3 and post-operative clinical outcomes	Cardioprotective effects of curcumin may include inhibition of the c-Jun N-terminal kinase pathway and caspase-3 in cardiomyocytes
Elbarbary, 2018 [134]	Egypt	90 patients with diabetic nephropathy; 45 in carnosine suppl. group (20 ♂, mean age 12.4) and 45 children in placebo group	Carnosine and urinary albumin excretion, alpha 1-microglobulin, oxidative stress	Oral supplementation with L-Carnosine for 12 weeks resulted in a significant improvement of oxidative stress, glycemic control and renal function
Xia, 2003 [135]	China	20 children with congenital heart defects and pulmonary hypertension, aged 2 to 15 years	Placebo or *Salvia miltiorrhiza* before cardiac surgery	Antioxidant therapy reduces myocardial damage and attenuates postoperative vasoactive mediator imbalance
Cherukuri, 2020 [136]	USA	67 individuals (26 ♂, mean age 59 years)	Plant based bioequivalent nitrate complex and NO in saliva	Endothelial function improved with reducing both systolic and diastolic BP in hypertensive individuals with daily supplementation of dietary NO_3_^−^
Hegazy, 2013 [137]	Egypt	30 T1DM patients, aged 10 to 14 years	Insulin vs. insulin plus alpha-lipoic acid	Alpha-lipoic acid may have a role in preventing the development of diabetic cardiomyopathy in T1DM
Scaramuzza, 2015 [138]	Italy	71 children and adolescents with T1DM, mean age 17 years	Antioxidant diet, alpha-lipoic acid, placebo	Alpha-lipoic acid might have an antioxidant effect in pediatric diabetes patients by reducing insulin
Rimoldi, 2015 [139]	Switzerland	21 children conceived by assisted reproductive technology (8 ♂, mean age 12) and 21 controls	Vitamin C and placebo	Antioxidant administration to participants improved NO bioavailability and vascular responsiveness in the systemic and pulmonary circulation
Roszkowska, 2014 [140]	Poland	87 babies and children, aged 5 to 32 months	Monocyte chemoattractant protein-1 and breast fed vs. bottle fed	Increased circulating inflammatory markers may indicate that formula-fed children are at risk of atherosclerosis
Bernini, 2016 [141]	Brazil	51 participants with MetS, aged 18 to 60 years	Probiotics and placebo	Cardiovascular risk in patients with MetS might be reduced by probiotics (*B. lactis* HN019)
Gobel, 2012 [142]	Denmark	50 adolescents with obesity, aged 12 to 15 years	Probiotic strain *Lactobacillus salivarius* and placebo	No beneficial effect of the probiotic intervention on inflammatory markers or parameters related to the MetS
Bresolin, 2013 [143]	Brazil	33 children with congenital heart disease, aged 7 to 12 years	CV risk parameters and dental treatment (scaling and root planing or full-mouth scaling and root planing)	Both periodontal treatments were effective in children with congenital heart disease in reduction of CV markers
Zeigler, 2015 [144]	Sweden	75 obese adolescents, aged 12 to 18 years	Obesity, periodontal disease and CV risk factors	Association between pathological periodontal pockets and diastolic BP
**Reviews and editorials**				
Berrahmoune, 2005 [1]	France	Human	Analysis of environmental, genetic and epigenetic determinants that influence inflammation in cardiovascular disease	Intermediate phenotype (protein and RNA level) variations in inflammation influence inter-individual risk of CVD
Barton, 2010 [2]	Switzerland	Human	Markers of oxidative stress and antioxidant capacity in obesity and aging and effect on vascular changes	Obesity and aging share similarities of pathomechanisms on vascular changes related to oxidative stress
Parisi, 2021 [145]	Italy	Animal and human	Intrauterine exposure to maternal low-grade chronic inflammation	Maternal obesity affects fetal programming of cardiovascular disease into adulthood
Skilton, 2008 [146]	France	Human	Intrauterine risk factors for precocious atherosclerosis	Impaired fetal growth, in utero exposure to maternal hypercholesterolemia and diabetic macrosomia as risk factors for fetal vascular changes
Leduc, 2010 [147]	Canada	Human	Fetal programming of atherosclerosis	Chronic in utero exposure to oxidative stress and inflammation, as also mitochondrial dysfunction may influence fetal programming of atherosclerosis
Silvestro, 2020 [148]	Italy	Animal and human	Role of placental oxidative stress on fetal development	Oxidative stress, mediated via placental hypoxia plays a key role in adverse effects on the developing offspring
Sutherland, 2014 [149]	Canada	Animal and human	Preterm birth, oxidative stress and hypertension risk	Oxidative stress associated with preterm birth affects neonatal development and future disease risk
Wang, 2015 [150]	Taiwan	Animal and human	Oxidative stress and renal damage in neonates	Recent studies elucidate the role of oxidative stress in neonatal renal damage
Yzydorczyk, 2017 [151]	Switzerland and France	Animal and human	Intrauterine growth restriction, endothelial dysfunction, CVD and renal consequences	Individuals born after IUGR exhibit vascular dysfunction, and preventive approaches (including antioxidant use) may mitigate long-term consequences
Hsu, 2020 [152]	Taiwan	Animal and human	Use of natural antioxidants as reprogramming strategies to precent developmental hypertension	Pregnant mothers and their children can benefit from natural antioxidant supplementation during pregnancy to reduce their risk for hypertension later in life
Radomski, 1995 [153]	Canada	Animal and human	The role of nitric oxide in the pathogenesis of atherosclerosis	It is possible that some of the detrimental effects of atherosclerosis on the nitric oxide pathway result from the generation of secondary oxidants such as peroxynitrite
Sabri, 2019 [154]	Iran	Human	Review of the epidemiology and risk factors in pediatric essential hypertension	Endothelial nitric oxide gene polymorphism is a possible factor to contribute to developing essential hypertension
Wirix, 2015 [155]	Netherlands	Human	Pathophysiology of hypertension in overweight and obese children	Development of hypertension in obese children is influenced by endocrine determinants, sympathetic nervous system activity, disturbed sodium homeostasis, as well as oxidative stress, inflammation and endothelial dysfunction
Filgueiras, 2020 [156]	Brazil	Human	Vitamin D status, oxidative stress and inflammation in children and adolescents	Vitamin D status is associated with oxidative stress and inflammation
Vincent, 2007 [157]	USA	Human	Oxidative stress and potential interventions to reduce oxidative stress in overweight and obesity	Oxidative stress may be corrected by improving antioxidant defenses through fat volume reduction via surgery, pharmacological agents, exercise and/or dietary modification
Espinola-Klein, 2011 [158]	Germany	Human	Inflammatory markers and cardiovascular risk in MetS	Elevations of inflammatory biomarkers impact the development of MetS and CVD
Balat, 2010 [159]	Turkey	Human	Nitric oxide (NO), adrenomedullin (AM), urotensin-II (U-II) in several pediatric renal diseases	NO, AM, and U-II may be important mediators in kidney diseases. They suggest a possible role in the pathophysiology of childhood glomerulonephritis
Massy, 2005 [160]	France	Human	Inflammation, oxidative stress, vascular calcifications in CKD	Inflammation and oxidative stress represent new features of the arterial and/or valvular calcification process
Avloniti, 2017 [161]	Greece	Animal and human	Oxidative stress responses to acute and chronic exercise in youth	Children and adolescents exhibit positive adaptations of their antioxidant system with exercise
Montero, 2012 [162]	France	Animal and human	Endothelial function, inflammation and oxidative stress in obese children	Obesity leads to endothelial dysfunction in children; early signs of CVD should be sought
Chen, 2012 [163]	Taiwan	Animal and human	Melatonin as antioxidant with anti-inflammatory effect	Melatonin modulates inflammation and increase mitochondrial biogenesis; potential therapeutic tool in a wide range of childhood disorders
Gombert, 2021 [164]	Spain	Animal and human	Melatonin and breast milk, antioxidant capacity, influence on weight gain	Melatonin has numerous benefits and is present in breast milk, making breastfeeding beneficial to long-term cardiovascular health, with an epigenetic effect on obesity

Abbreviations: BP, blood pressure. CKD, chronic kidney disease. CRF, chronic kidney failure. CRP, C-reactive protein. CVD, cardiovascular disease. ESRD, end stage renal disease. HD, hemodialysis. IMT, intima-media thickness. IUGR, intrauterine growth restriction. IVF, in-vitro fertilization. LVH, left ventricular hypertrophy. MetS, Metabolic Syndrome. MI, myocardial infarction. NA, not available. NO, nitic oxide. OSA, obstructive sleep apnea. RANTES, Regulated upon Activation, Normal T Cell Expressed and Presumably Secreted. T1DM, type 1 diabetes mellitus. T2DM, type 2 diabetes mellitus. UK, United Kingdom. USA, United States of America.

**Table 2 antioxidants-11-00894-t002:** Risk factors and diseases associated with oxidative stress, systemic inflammation and cardiovascular disease.

Risk Factors and Diseases	
Genetic predisposition	nitric oxide synthase gene polymorphism
	gene polymorphisms of the renin–angiotensin–aldosterone system and aldosterone synthase genes
	single-nucleotide polymorphism of ATP2B1 (ATPase Plasma Membrane Ca^2+^ Transporting 1) familial hypercholesterolemia
Perinatal factors	conception by assisted reproductive technology maternal hypercholesterolemia, obesity, metabolic syndrome placental insufficiency being small for gestational age preterm delivery formula-feeding
Environmental factors	lack of exercise
	poor nutrition
	exposure to phthalates
	lower socioeconomic status
Psycho-emotional factors	anxiety irritability predisposition to depressive reactions
Weight status	Underweight Overweight Obese
Immunosuppressive treatment	cyclosporine tacrolimus
Dialysis	
Hypovitaminosis	D and B12
Hyperhomocysteinemia	
Kidney diseases	chronic kidney disease hydronephrosis glomerulonephritis pyelonephritis lower urinary tract infections
Arterial hypertension	essential and secondary
Obstructive sleep apnea	
Diabetes mellitus	type 1 and 2
Metabolic syndrome	

**Table 3 antioxidants-11-00894-t003:** Oxidative and inflammatory markers associated with cardiovascular disease.

Oxidative Stress Markers	Antioxidant System Markers	Adipokines and Other Systemic Inflammation Markers
**Lipid peroxidation** F_2_ isoprostanes malondialdehyde (MDA) thiobarbituric acid reactive substances (TBARS) oxidized low density lipoprotein (oxLDL) **Protein oxidation** Advanced oxidation protein products (AOPPS) **Carbohydrate oxidation** Advanced glycosylation end-products (AGEs) **Nucleic acid oxidation** 8-hydroxy-2′-deoxyguanosine (8-OHdG) **Reactive oxygen species (ROS) generation** myeloperoxidase (MPO) NADPH ^1^ oxidase (NOX2) **Nitric oxide system (NOx)** polyamines derived from arginine asymmetric dimethyloarginine (ADMA) nitrite and nitrate NO	Thiol/disulphide homeostasis glutathione (GSH) superoxide dismutase (SOD) catalase (CAT) glutathione peroxidase (GPx) carotenes (vitamin A) ascorbic acid (vitamin C) tocopherols (vitamins E) bilirubin ceruloplasmin Total antioxidant capacity (TAC)	Chemerin adiponectin leptin resistin visfatin adipomyokine irisin RANTES ^2^ monocyte chemoattractant protein-1 (MCP-1) stromal-derived factor (SDF-1) interleukins IL-1, -1β, -6, -10, -18 tumor necrosis factor alpha (TNF-α) plasminogen activator-inhibitor-1 (PAI-1) α-1-acid glycoprotein (AGP) high sensitivity C-reactive protein (hsCRP) C-reactive protein (CRP) myeloperoxidase (MPO)

^1^ NADPH, nicotinamide-adenine dinucleotide phosphate oxidase. ^2^ RANTES, Regulated upon Activation, Normal T Cell Expressed and Presumably Secreted.

## Data Availability

All the data are available within the article.

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
