# Peer review of "Systemic Inflammation, Oxidative Stress and Cardiovascular Health in Children and Adolescents: A Systematic Review"

_antioxidants, 2022, doi:10.3390/antiox11050894_

Round 1

Reviewer 1 Report

The manuscript by Petek et al. summarizes the evidence describing the role of systemic inflammation and oxidative stress in cardiovascular health in chronic pediatric conditions. The authors have done a thorough literature search to identify relevant articles. However, the article is verbose. For clarity, authors should list the markers, risk factors and diseases in a tabular format. 

Author Response

Dear reviewer,

Thank you for revising our manuscript.

The answers to your comments are given below.

The manuscript by Petek et al. summarizes the evidence describing the role of systemic inflammation and oxidative stress in cardiovascular health in chronic pediatric conditions. The authors have done a thorough literature search to identify relevant articles. However, the article is verbose. For clarity, authors should list the markers, risk factors and diseases in a tabular format.

  • We have included a list of markers, risk factors and diseases in a tabular format.

Thank you again for taking your valuable time to improve our manuscript.

Kind regards,

The authors

Reviewer 2 Report

  1. Introduction- perhaps there are length limitations, but to strength the argument for the impact of ROS on health for children, I think a minor addition discussion oxidized LDL as a potential etiology for the atherosclerotic cascade would be beneficial to some of the readers.
  2. I have no issues with the methodology, this seems standard.
  3. Regarding genetic factors I think an interesting addition could be made with the potential polymorisms of necessary enzymes in antioxidant pathways, like GCLC and the relation with heart disease.
    1. Koide, S. I., Kugiyama, K., Sugiyama, S., Nakamura, S. I., Fukushima, H., Honda, O., ... & Ogawa, H. (2003). Association of polymorphism in glutamate-cysteine ligase catalytic subunit gene with coronary vasomotor dysfunction and myocardial infarction. Journal of the American College of Cardiology41(4), 539-545.
    2. A lot could be added about the antioxidant and fitness status of the parent/egg/sperm cells. (But I don’t think it is exactly necessary)

4.2- This section appears to need slightly more discussion of adiponectin.

4.5- Please revise this sentence “Anyway, in children requiring hemodialysis for several years, coronary calcifications are common, especially in association with worse osteodystrophy control and malnutrition.” It is somewhat unclear what is being said. Line 405.

4.7- This is true about what type 1 diabetes is; however, I think the autoimmune issues with the beta cells is a better starting point for most readers.

  1. Granted this is outside the topic of children since this is normally not a recommendation so it is up to the authors discretion to include this, but there is indication that carbohydrate restriction might alter total oxidative stress in a positive direction.

Greco, T., Glenn, T. C., Hovda, D. A., & Prins, M. L. (2016). Ketogenic diet decreases oxidative stress and improves mitochondrial respiratory complex activity. Journal of Cerebral Blood Flow & Metabolism36(9), 1603-1613.

The authors could also look into methionine restriction if word counts aren’t exceeded.

I find the rest of the paper to be very well written and thought out.

Author Response

Dear reviewer,

Thank you for revising our manuscript.

The answers to your comments are given below.

Introduction- perhaps there are length limitations, but to strength the argument for the impact of ROS on health for children, I think a minor addition discussion oxidized LDL as a potential etiology for the atherosclerotic cascade would be beneficial to some of the readers.

  • An introductory sentence has been added on the importance of oxidized LDL in the pathogenesis of atherosclerosis.

I have no issues with the methodology, this seems standard.

Regarding genetic factors I think an interesting addition could be made with the potential polymorisms of necessary enzymes in antioxidant pathways, like GCLC and the relation with heart disease.

  • This has been included in the genetic risk section of the manuscript.

Koide, S. I., Kugiyama, K., Sugiyama, S., Nakamura, S. I., Fukushima, H., Honda, O., ... & Ogawa, H. (2003). Association of polymorphism in glutamate-cysteine ligase catalytic subunit gene with coronary vasomotor dysfunction and myocardial infarction. Journal of the American College of Cardiology, 41(4), 539-545.

A lot could be added about the antioxidant and fitness status of the parent/egg/sperm cells. (But I don’t think it is exactly necessary)

- To omit excessive word count, this has not been included.

4.2- This section appears to need slightly more discussion of adiponectin.

- A paragraph has been added.

4.5- Please revise this sentence “Anyway, in children requiring hemodialysis for several years, coronary calcifications are common, especially in association with worse osteodystrophy control and malnutrition.” It is somewhat unclear what is being said. Line 405.

- This sentence has been revised and is now clarified.

4.7- This is true about what type 1 diabetes is; however, I think the autoimmune issues with the beta cells is a better starting point for most readers.

- The autoimmune pathogenesis of type 1 diabetes is now highlighted.

Granted this is outside the topic of children since this is normally not a recommendation so it is up to the authors discretion to include this, but there is indication that carbohydrate restriction might alter total oxidative stress in a positive direction.

  • Now included in the »exercise and diet« section.

Greco, T., Glenn, T. C., Hovda, D. A., & Prins, M. L. (2016). Ketogenic diet decreases oxidative stress and improves mitochondrial respiratory complex activity. Journal of Cerebral Blood Flow & Metabolism, 36(9), 1603-1613.

The authors could also look into methionine restriction if word counts aren’t exceeded. I find the rest of the paper to be very well written and thought out.

  • Now also included in the »exercise and diet« section.

Thank you again for taking your valuable time to improve our manuscript.

Kind regards,

The authors

Reviewer 3 Report

Comments to the Author

In this paper, the authors aimed to assess the potential systemic inflammation, oxidative stress, and antioxidant use, endothelial dysfunction for cardiovascular health in children and adolescents by literature review. This study is some interesting and the results may be useful. However, some critical concerns should be addressed before publish.

1- I suggested the authors should register this review to PROSPERO.

2- The quality of prisma flow chart is satisfied. Besides, I suggested the authors should add some published article for this meta-analysis if necessary.

3- To our knowledge, prisma2020 was updated since 2021. What the difference between prisma2009 and prisma2020? Maybe conduct a sensitive analysis for included studies in this review.

4- Where is the mechanism plot described in manuscript based on its assumed pathogenesis? This concern needs to be addressed.

5- To my knowledge, discussion should updated solution or treatment for this issue. Could the authors conduct related treatment for alleviation of potential systemic inflammation, oxidative stress, and antioxidant use, endothelial dysfunction associated with cardiovascular disease?

Author Response

Dear reviewer,

Thank you for revising our manuscript.

The answers to your comments are given below.

In this paper, the authors aimed to assess the potential systemic inflammation, oxidative stress, and antioxidant use, endothelial dysfunction for cardiovascular health in children and adolescents by literature review. This study is some interesting and the results may be useful. However, some critical concerns should be addressed before publish.

1- I suggested the authors should register this review to PROSPERO.

  • We have submitted a systematic review registration to PROSPERO. This is now added in the Methods section.

2- The quality of prisma flow chart is satisfied. Besides, I suggested the authors should add some published article for this meta-analysis if necessary.

  • To our best knowledge, no meta-analysis of oxidative stress, systemic inflammation and cardiovascular health & disease in the pediatric population has been published to date.

3- To our knowledge, prisma2020 was updated since 2021. What the difference between prisma2009 and prisma2020? Maybe conduct a sensitive analysis for included studies in this review.

  • We used the latest PRISMA 2020 Flow Diagram for »new systematic reviews which included searches of databases and registers only«, published in 2021,. already in the first version of the manuscript. We retrieved the diagram from the PRISMA webpage: http://prisma-statement.org/prismastatement/flowdiagram.aspx. As per Matthew J Page (doi: https://doi.org/10.1136/bmj.n71) , »the PRISMA 2020 statement replaces the 2009 statement and includes new reporting guidance that reflects advances in methods to identify, select, appraise, and synthesize studies. The structure and presentation of the items have been modified to facilitate implementation.«
  • Due to a large heterogeneity of studies in this systematic review, based on varied population, interventions, and varied outcomes, we concluded that a formal sensitivity analysis would not be feasible, nor would it contribute to greater robustness of the findings.

4- Where is the mechanism plot described in manuscript based on its assumed pathogenesis? This concern needs to be addressed.

  • The pathogenetic background on the effects of systemic inflammation and oxidative stress on endothelial cells, leading to vascular dysfunction and cardiovascular disease are now explained in greater detail.

5- To my knowledge, discussion should updated solution or treatment for this issue. Could the authors conduct related treatment for alleviation of potential systemic inflammation, oxidative stress, and antioxidant use, endothelial dysfunction associated with cardiovascular disease?

  • We have added an additional two paragraphs in the conclusion section, summarizing studies with positive results on treatment with antioxidants in the pediatric population at risk for or with already present cardiovascular disease.

Thank you again for taking your valuable time to improve our manuscript.

Kind regards,

The authors

Reviewer 4 Report

This is a review article, regarding systemic inflammation, oxidative stress and cardiovascular health in children and adolescents. The authors well summarized genetic and perinatal factors affecting oxidative stress and systemic inflammation, chronic diseases on cardiovascular health via oxidative stress and systemic inflammation, influence of exercise and diet, and others. This is an important issue, and this reviewer considers that the authors well written this review article, and has some comments as described below.

Major comments:

  1. The authors mentioned about vitamins or natural antioxidants, which were anti-oxidant side, but they did not focus on unhealthy food intake, which worsen oxidative state. The authors should describe also about unhealthy food.
  2. Subheading 5 and 6. They were completely the same. Section 5 seems to be about exercise, and section 6 may be about diet. Their subheadings should be different.

Author Response

Dear reviewer,

Thank you for revising our manuscript.

The answers to your comments are given below.

This is a review article, regarding systemic inflammation, oxidative stress and cardiovascular health in children and adolescents. The authors well summarized genetic and perinatal factors affecting oxidative stress and systemic inflammation, chronic diseases on cardiovascular health via oxidative stress and systemic inflammation, influence of exercise and diet, and others. This is an important issue, and this reviewer considers that the authors well written this review article, and has some comments as described below.

Major comments:

The authors mentioned about vitamins or natural antioxidants, which were anti-oxidant side, but they did not focus on unhealthy food intake, which worsen oxidative state. The authors should describe also about unhealthy food.

  • We have included a paragraph on unhealthy food intake and oxidative stress at the end of subheading 5.

Subheading 5 and 6. They were completely the same. Section 5 seems to be about exercise, and section 6 may be about diet. Their subheadings should be different.

  • The duplicate subheading has been noted and removed. Subheading 6 has been changed to an appropriate name.

Thank you again for taking your valuable time to improve our manuscript.

Kind regards,

The authors

Reviewer 5 Report

Ambitious and well presented and condensed review. from my point as interested in socioeconomic and educational effects - i miss  a paragraph on this.

Vitamins K1/K2 are in focus with effects on vascular mediasclerosis and MGP (dp-ucMGP, Gas6 and osteocalcin (antidiabetic antilipid effects)). Vitamin E counteracts vitamin K effects.

What to measure: frequencies of Stroke, MI, vascular stiffness, infection rate?

Is there an accepted WHO parameter on health?

Specific -nut intake and other nutrient effects on immune system ("immune nutritrion") and brain ("brain food").

Type of exercise (static upper or lower body exercises, running, lifting weights, etc - much research is ongoing on effects on vascular stiffness.

Should children be medicated early with statins/antidiabetics or even undergo obesity surgery (much research indicate the latter)

White people (caucasians) contra asian/african? Genetics involved in differences between these groups on top of nutrition?

Author Response

Dear reviewer,

Thank you for revising our manuscript.

The answers to your comments are given below.

Ambitious and well presented and condensed review. from my point as interested in socioeconomic and educational effects - i miss  a paragraph on this.

A paragraph has been added.

Vitamins K1/K2 are in focus with effects on vascular mediasclerosis and MGP (dp-ucMGP, Gas6 and osteocalcin (antidiabetic antilipid effects)). Vitamin E counteracts vitamin K effects.

Thank you for the add-on, it has been implemented in the appropriate section with an appropriate reference.

What to measure: frequencies of Stroke, MI, vascular stiffness, infection rate?

We would say, all of them, if possible. However, as stroke and MI are very uncommon in the pediatric population, vascular stiffness is one of the most important cardiovascular risk markers in children and adolescents.

Is there an accepted WHO parameter on health?

This is an interesting question. After research, no one specifically can be identified, however, as in our review, WHO concentrates more on disease tracking (in WHO, epidemiologically, in our research in individual from birth on) than health tracking. Health parameters should be researched and defined further, as with health definition itself.

Specific -nut intake and other nutrient effects on immune system ("immune nutritrion") and brain ("brain food").

A paragraph has been added.

Type of exercise (static upper or lower body exercises, running, lifting weights, etc - much research is ongoing on effects on vascular stiffness.

 – A paragraph has been added.

Should children be medicated early with statins/antidiabetics or even undergo obesity surgery (much research indicate the latter)

Medications and sugeries due to the obesity are generally discouraged in pediatric population due to long-term side effects. Also, if there is no lifestyle change, medications do not help significantly, nor does surgery prevents further cardiovascular complicatins and possible additional weight gain. A paragraph on the topic has been added.

White people (caucasians) contra asian/african? Genetics involved in differences between these groups on top of nutrition?

A paragraph on ethnic differences in atherosclerosis and cardiovascular risk has been added.

Thank you again for taking your valuable time to improve our manuscript.

Kind regards,

The authors

Round 2

Reviewer 3 Report

Comments to the Author

Thanks for your efforts on revision. I had made concerns according to your response.

1- Please provide registered code of PROSPERO.

2- No further comments. But, published meta-analysis about treatment of makers for cardiovascular disease should be added in discussion section. This concern supplemented previous comment #5. For example, PMID: 31487802.  

3- Satisfied with your response.

4- No further comments.

5- The authors should cite appropriate references corresponding to the specific treatments for alleviation of potential systemic inflammation, oxidative stress, and antioxidant use, endothelial dysfunction associated with cardiovascular disease. Thanks for previous response.

Author Response

Dear reviewer,

Thank you for revising our manuscript.

The answers to your comments are given below.

1- Please provide registered code of PROSPERO.

                The study is undergoing registration by PROSPERO (record ID: 316807) for more than one month. We have recieved notice from PROSPERO that for submissions outside the UK, such as ours, a response may take up to three months and during this time we may continue working on our manuscript.

2- No further comments. But, published meta-analysis about treatment of makers for cardiovascular disease should be added in discussion section. This concern supplemented previous comment #5. For example, PMID: 31487802.

                We have included three new review articles on therapeutic interventions in the discussion section, which provide an oversight of the topic.

3- Satisfied with your response.

4- No further comments.

5- The authors should cite appropriate references corresponding to the specific treatments for alleviation of potential systemic inflammation, oxidative stress, and antioxidant use, endothelial dysfunction associated with cardiovascular disease. Thanks for previous response.

                In the revised manuscript we have now included five new references (182-186) which review systemic inflammation, oxidative stress, endothelial dysfunction and specific antioxidant therapeutic approaches, aimed at improving cardiovascular disease markers. These are mixed studies on adult and pediatric populations.

This is followed in the discussion section by studies on more specific beneficial uses of supplementation with vitamin C, E, zinc, grape and pomegranate juice, carnosine, herb extracts and phytophenols, alpha-lipoic acid, as well as probiotics, which we already cited in the Sections 3 to 7 of the manuscript. In the Section 8 – Conclusions and future perspectives, we thus provide a synthesis of aforementioned study results.

Thank you again for taking your valuable time to improve our manuscript.

Kind regards,

Tadej Petek on behalf of the co-authors